# Dissecting the Interplay of Attention Paths in a Statistical Mechanics Theory of Transformers

**Lorenzo Tiberi**[1,2]   **Francesca Mignacco**[3,4]   **Kazuki Irie**[1,2]   **Haim Sompolinsky**[1,2,5]

[1]Center for Brain Science, Harvard University, Cambridge, MA, USA
[2]Kempner Institute for the Study of Natural and Artificial Intelligence,
Harvard University, Cambridge, MA, USA
[3]Graduate Center, City University of New York, NY, USA
[4]Joseph Henry Laboratories of Physics, Princeton University, NJ, USA
[5]Edmond and Lily Safra Center for Brain Sciences,
Hebrew University of Jerusalem, Jerusalem, Israel
`ltiberi@fas.harvard.edu, fmignacco@princeton.edu`
`kirie@fas.harvard.edu, hsompolinsky@mcb.harvard.edu`

## Abstract

Despite the remarkable empirical performance of transformers, their theoretical understanding remains elusive. Here, we consider a deep multi-head self-attention network, that is closely related to transformers yet analytically tractable. We develop a statistical mechanics theory of Bayesian learning in this model, deriving exact equations for the network's predictor statistics under the finite-width thermodynamic limit, i.e., $N, P \to \infty$, $P/N = \mathcal{O}(1)$, where $N$ is the network width and $P$ is the number of training examples. Our theory shows that the predictor statistics are expressed as a sum of independent kernels, each one pairing different *attention paths*, defined as information pathways through different attention heads across layers. The kernels are weighted according to a *task-relevant kernel combination* mechanism that aligns the total kernel with the task labels. As a consequence, this interplay between attention paths enhances generalization performance. Experiments confirm our findings on both synthetic and real-world sequence classification tasks. Finally, our theory explicitly relates the kernel combination mechanism to properties of the learned weights, allowing for a qualitative transfer of its insights to models trained via gradient descent. As an illustration, we demonstrate an efficient size reduction of the network, by pruning those attention heads that are deemed less relevant by our theory.[1]

## 1  Introduction

In recent years, transformer models based on multi-head self-attention layers [1–6] have achieved remarkable performance at natural language processing and vision tasks [7–9]. Yet, theoretical characterizations accounting for the success of these architectures remain sparse. Two fundamental questions remain to a large extent unsolved: First, interpretability—how can we discern task-relevant structures within the learned weights? Second, generalization—what specific aspects of the transformer architecture are responsible for their effective learning? We posit that one important feature of transformers is the combination of layer-wise multi-head organization with depth. This provides the network with a large number of *attention paths*, defined as specific sequences of heads through the attention layers. Their *interplay* is still poorly understood by deep learning theory.

---

[1]Our code is public: https://github.com/tiberilor/attention-paths-interplay

38th Conference on Neural Information Processing Systems (NeurIPS 2024).

In most cases, theoretical characterizations of transformers' expressivity [10], inductive bias [11, 12], generalization [13–15] and training dynamics [16–18] rely on simplifying assumptions on the network architecture. A characterization of attention paths is inaccessible in these models, either because attention paths are not defined in the first place, as in models consisting of a single-head [19, 15], a single-layer [10, 11], or both [12–14, 16, 18, 20], or because the interplay between paths cannot be fully described due to constraints imposed on the learnable weights [17]. A few works consider a multi-head, multi-layer architecture [21–24], but address different questions than the present study, such as expressivity, generalization bounds, or phenomenological models. Further details on these and analogous works are discussed in Appendix I.

One characterization of the complete transformer architecture has been obtained in the Bayesian framework under the *infinite-width* thermodynamic limit $N \to \infty$ (and infinite number of heads $H \to \infty$) [25, 26], an actively studied regime in which neural networks become equivalent to Gaussian processes (GP) [27, 28]. However, the attention paths interplay is lost in this limit because the network's hidden weights remain statistically independent after learning. This limitation can be overcome by considering the *finite-width* thermodynamic limit [29–32], where also the number of examples $P \to \infty$ such that $P/N \to \alpha \in \mathbb{R}^+$. In this regime, for example, multi-gated deep networks showcase task-relevant interplay between gates, mediated by the learned weights [33].

In this work, we apply the statistical mechanics theory of finite-width networks to a deep multi-head self-attention model, which closely mimics the attention paths interplay in transformers, while remaining analytically tractable. Our main contributions can be summarized as follows:

- We derive exact equations for the predictor statistics under Bayesian learning of the network's value weights, at fixed query and key weights.

- We shed light on the interplay between attention paths by uncovering a *task-relevant kernel combination* mechanism, emerging beyond the GP limit ($\alpha > 0$). This constructs the network's mean predictor as an optimally weighted sum of many "path-path kernels", defined as similarity matrices between pairs of attention paths, thereby improving generalization.

- We provide interpretability to this mechanism, by directly relating it to the magnitude and correlations developed by the learned weights. This allows our insights to be transferred outside the Bayesian framework, to networks trained with gradient descent. As an application, we show that a trained network can be reduced in size with minimal performance loss, by pruning those attention paths that are deemed less relevant by our theory.

- We corroborate our findings on both synthetic and real-world sequence classification tasks, illustrating the two main benefits of kernel combination: task-relevant weighting and correlation of the attention paths, respectively.

## 2 Model

We consider a transformer-like [1] architecture consisting of a linear input projection layer; $L$ multi-head self-attention (MHA) layers, each having $H$ attention heads; and a linear readout layer. The network input $x \in \mathbb{R}^{N_0 \times T}$ is a sequence of $T$ tokens $x_t \in \mathbb{R}^{N_0}$, with token index $t \in \{1, \ldots, T\}$, and dimension $N_0$. The input projection layer performs the transformation

$$x_t^{(1)} = \frac{1}{\sqrt{N_0}} V^{(0)} \cdot x_t, \qquad V^{(0)} \in \mathbb{R}^{N \times N_0}, \tag{1}$$

where $N$ is the hidden layers' width. With the operator "·" we denote matrix-matrix or matrix-vector multiplication. The $\ell$-th MHA layer with index $\ell \in \{1, \ldots, L\}$ performs the transformation

$$x_t^{(\ell+1)} = \frac{1}{\sqrt{NH}} \sum_{h=1}^{H} \sum_{s=1}^{T} V^{(\ell)h} \cdot x_s^{(\ell)} \Omega_{st}^{(\ell)h}, \qquad V^{(\ell)h} \in \mathbb{R}^{N \times N}, \tag{2}$$

where, for each head $h$, we define the attention matrix $\Omega^{(\ell)h} \in \mathbb{R}^{T \times T}$ with matrix elements

$$\Omega_{st}^{(\ell)h} = \zeta \left( \frac{1}{N_0 \sqrt{G}} x_s^\top \cdot W_K^{(\ell)h\top} \cdot W_Q^{(\ell)h} \cdot x_t \right), \qquad W_Q^{(\ell)h}, W_K^{(\ell)h} \in \mathbb{R}^{G \times N_0}. \tag{3}$$

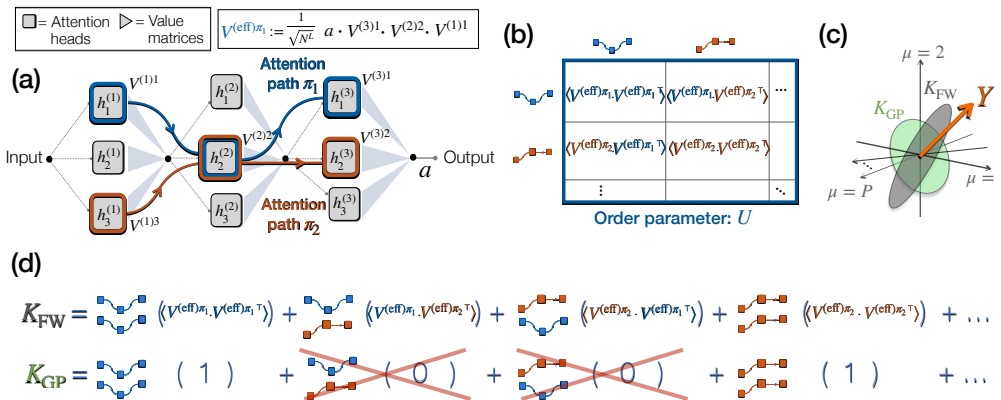

Figure 1: **Scheme of the model and theory (a)** Scheme of the model in terms of attention paths. **(b)** The order parameter assigns to each pair of paths a weight, given by the overlap between the corresponding effective weights. **(c)** Alignment of the kernel PCs with the vector of task labels $Y$, in the finite-width (FW) vs GP regimes. **(d)** Kernel as the weighted sum of many path-path kernels. Task-relevant kernel combination occurs in the finite-width regime (FW), but not in the GP limit, in which cross-path kernels are discarded, and same-path kernels are equally weighted. The result is an improved kernel-task alignment in the finite-width regime (shown in (c)), enhancing generalization.

Here $\zeta$ is the softmax function, applied along the direction of the token index $s$, while $G$ is the dimension of the query-key feature space. The linear readout returns the scalar output

$$f = \frac{1}{\sqrt{N}} a \cdot x_{t^*}^{(L+1)}, \qquad a \in \mathbb{R}^{1 \times N}. \tag{4}$$

Here $x_{t^*}$ can stand for different options for reducing the token dimension at readout, namely reading from a specific token $t^*$ or averaging over all tokens ($x_{t^*} := \frac{1}{T} \sum x_t$). The network's learnable parameters are the input projection weights $V^{(0)}$; the value, query and key weights $\left\{ V^{(\ell)h}, W_Q^{(\ell)h}, W_K^{(\ell)h} \right\}_{\ell,h=1}^{L,H}$; and the readout weights $a$.

**Comparison to the Standard Transformer.** The above architecture presents two main simplifications w.r.t. the standard transformer. First, the network is linear in the value weights, while the standard transformer has a nonlinear feedforward block after each MHA layer. Second, in any layer $\ell$, the attention (Eq. 3) is always computed as a direct function of the bare input $x$, rather than the processed input $x^{(\ell)}$. These simplifications allow us to apply back-propagating kernel renormalization (BPKR) techniques [29, 33], enabling the characterization of the network beyond the GP limit. Despite these simplifications, the insights gained by going beyond the GP limit are substantial: we will show that, in the finite-width regime, an important mechanism—*task-relevant kernel combination*—emerges, accounting for a considerable improvement in generalization performance.

**Attention paths formulation.** Note that, despite the linearization in the value weights, the network is still highly nonlinear in the input, thanks to the attention operation (Eq. 3). This can be seen by the following equivalent description of the network (Fig. 1(a)). We introduce the concept of *attention paths*, by defining a path "index" $\pi := (h_1, h_2, \ldots, h_L)$, where $h_1, \ldots, h_L \in \{1, \ldots, H\}$, which uniquely identifies each possible combination of the head indices across layers, i.e., each possible path through the attention heads. The network output can be rewritten as

$$f = \frac{1}{\sqrt{H^L N N_0}} \sum_{\pi \in \Pi} V^{(\text{eff})\pi} \cdot V^{(0)} \cdot \xi^\pi \tag{5}$$

where $\Pi$ is the set of all possible paths, and we define the "*effective weights*" as

$$V^{(\text{eff})\pi} := \frac{1}{\sqrt{N^L}} a \cdot V^{(L)h_L} \cdot V^{(L-1)h_{L-1}} \cdot \ldots \cdot V^{(2)h_2} \cdot V^{(1)h_1}, \qquad V^{(\text{eff})\pi} \in \mathbb{R}^{1 \times N} \tag{6}$$

and the "*attentioned input*" as

$$\xi^\pi := \sum_{t_0,\ldots,t_{L-1}=1}^{T} x_{t_0} \Omega_{t_0 t_1}^{(1)h_1} \Omega_{t_1 t_2}^{(2)h_2} \ldots \Omega_{t_{L-2} t_{L-1}}^{(L-1)h_{L-1}} \Omega_{t_{L-1} t^*}^{(L)h_L}, \qquad \xi^\pi \in \mathbb{R}^{N_0}. \tag{7}$$

In Eq. (5), the network can be seen as a deep linear network applied to a nonlinearly expanded input—the attentioned input $\xi^{\pi}$. Through Eq. (7), we can see that the bare input $x$ is nonlinearly expanded from an $N_0$-dimensional space to an $N_0 H^L$-dimensional space, by means of $H^L$ nonlinear operations: one for each attention path.

The goal of our theory is to understand how the network learns to combine these different attention paths, by means of the effective weights (Eq. 6). Note that the network also has other learnable parameters: the query and key weights, which parameterize the nonlinear expansion of the input to $\xi$. The learning of these parameters is not described by our theory. As we will see in Sec. 3, our theory characterizes the learned effective weights (Eq. 6) for a given, fixed realization of the query and key weights.

## 3 Theory

A fundamental quest of deep learning theory is to understand how deep neural networks, which are often overparameterized, manage to avoid overfitting, achieving good generalization performance [34, 35]. One important role is played by the specific choice of network architecture, which can impose an inductive bias towards better generalizing configurations of parameters, among the many that fit the training data. To study this problem, we adopt the Bayesian framework. Given a dataset of $P$ example-label pairs $\{x^{\mu}, y^{\mu}\}_{\mu=1}^{P}$, we seek to characterize the Bayesian posterior [36–38], or *Gibbs distribution*, over the parameters $\Theta := \left( V^{(0)}, \{V^{(\ell)h}\}_{\ell,h=1}^{L,H}, a \right)$

$$p\left(\Theta\right) \propto \exp\left\{ -\frac{1}{2\mathcal{T}} \sum_{\mu=1}^{P} \left[f\left(x^{\mu}, \Theta\right) - y^{\mu}\right]^2 - \frac{1}{2\sigma^2} \|\Theta\|^2 \right\}. \tag{8}$$

Here $f\left(x^{\mu}, \Theta\right)$ is the network output (Eq. 4) corresponding to the input $x^{\mu}$, where we emphasize its dependence on $\Theta$, $\|\cdot\|$ is the Frobenius norm, $\sigma^2$ is the variance of the weights' Gaussian prior (set to $\sigma = 1$ throughout this paper), and $\mathcal{T} > 0$ is the error variance, or *Gibbs temperature* (not to be confused with the number of tokens $T$). Characterizing the Gibbs distribution allows to gain insights into the inductive bias imposed by the network architecture. Indeed, note that, in overparameterized networks, the Gibbs distribution for $\mathcal{T} \to 0^+$ describes the statistics of those parameter configurations that perfectly fit the training data, with a bias towards small weights induced by the Gaussian prior. These statistics depend on the choice of network architecture, which can therefore bias the distribution towards better generalizing parameter configurations. For $\mathcal{T} > 0$, parameter configurations that do not achieve perfect fitting are also allowed, which can help to prevent overfitting.

Note that, as discussed at the end of Sec. 2, we characterize the statistics of the weights $\Theta$ (the linear projection, value, and readout weights) for a fixed realization of the query and key weights. The fixed query and key weights can be given, for example, by pre-training the network with gradient descent, or by some task-informed initialization. In Sec. 4.2 we will show that the insights gained by our theory on the weights $\Theta$ can also be applied to the network trained with gradient descent on all of its learnable parameters, including the query and key weights.

The main theoretical result of this work is an expression for the expectation $\mathbb{E}\left[f\left(x^*\right)\right]$ of the network's output on a new test example $x^*$, under the Gibbs distribution (Eq. 8). In Sec. 3.1 below, we provide this formal result, accompanied by a sketch of its derivation and a discussion of the significance of the infinite-dimensional—or thermodynamic— limit under which our result is derived. In Sec. 3.2 we discuss the result's interpretation and its insights into the network's generalization capabilities.

### 3.1 Statement of theoretical results

**Definitions**. Consider a training dataset consisting of $P$ inputs $x^{\mu} \in \mathbb{R}^{N_0 \times T}$ and associated labels $y^{\mu} \in \mathbb{R}$, where $\mu = 1, \ldots P$. Call $X := \{x^{\mu}\}_{\mu=1}^{P}$ the set of training inputs and $Y \in \mathbb{R}^P$ the vector of training labels with $\mu$-th component $y^{\mu}$. Consider a network defined by Eqs. (1-4) and in particular call $f^*$ the network output (Eq. 4) corresponding to a test input $x^* \in \mathbb{R}^{N_0 \times T}$.

**Assumptions**. Assume the query and key weights $\left\{W_Q^{(\ell)h}, W_K^{(\ell)h}\right\}_{\ell,h=1}^{L,H}$ are fixed, while all other weights $\Theta := \left( V^{(0)}, \{V^{(\ell)h}\}_{\ell,h=1}^{L,H}, a \right)$ are distributed according to the Bayesian posterior distribu-

tion defined in Eq. (8). Assume the "thermodynamic limit" $N, N_0, P \to \infty$, with $P/N := \alpha \in \mathbb{R}^+$ and $P/(N_0 H^L) := \alpha_0 \in \mathbb{R}^+$, where $\alpha, \alpha_0$ as well as other size parameters $T, H, L \in \mathbb{N}$ are finite.

**Result 1**. The mean predictor under the posterior distribution (Eq. 8) is given by

$$\mathbb{E}\left[f^*\right] = k^\top \cdot (K + \mathcal{T}\mathbb{I})^{-1} Y, \tag{9}$$

The vector $k \in \mathbb{R}^{P \times 1}$ and the matrix $K \in \mathbb{R}^{P \times P}$, called training kernel, are defined in terms of a kernel function $\mathcal{K} : \mathbb{R}^{N_0 \times T} \times \mathbb{R}^{N_0 \times T} \to \mathbb{R}$ as $k^\mu := \mathcal{K}(x^*, x^\mu)$ and $K^{\mu\nu} := \mathcal{K}(x^\mu, x^\nu)$, for $\mu, \nu = 1, \ldots, P$. The kernel function is given by

$$\mathcal{K}(x, x') = \frac{1}{H^L} \sum_{\pi, \pi' \in \Pi} U^{\pi\pi'} C_{\pi\pi'} \qquad \text{with} \qquad C_{\pi\pi'} := \frac{1}{N_0} \xi^\pi(x)^\top \cdot \xi^{\pi'}(x'), \tag{10}$$

where $\xi^\pi(x)$ is the "attentioned input" corresponding to an input $x \in \mathbb{R}^{N_0 \times T}$, along path $\pi \in \Pi$, as defined in Eq. (7). The kernel function depends on a positive semi-definite matrix $U \in \mathbb{R}^{H^L \times H^L}$, called *order parameter*, which is given by

$$U = \underset{\tilde{U}}{\operatorname{argmin}} \, S(\tilde{U}; X, Y) \qquad \text{with} \qquad S(U; X, Y) = -\mathcal{L}(U) + \alpha \mathcal{E}(U; X, Y), \tag{11}$$

The scalar function $S$, called the *action*, consists of an "*entropy*" term $\mathcal{L}$, and an "*energy*" term

$$\mathcal{E}(U; X, Y) = \frac{1}{P} \ln \det (K(U; X) + \mathcal{T}\mathbb{I}) + \frac{1}{P} Y^\top \cdot (K(U; X) + \mathcal{T}\mathbb{I})^{-1} \cdot Y, \tag{12}$$

where $K(U; X) := K$ is the training kernel matrix. The expression for the entropy $\mathcal{L}$ is lengthy and is given in Appendix B.1. In the special case of $H = 1$, $U$ is a scalar, and $\mathcal{L}(U) = -\sigma^{-2(L+1)} U + \ln(U)$. For general $H$, the entropy $\mathcal{L}(U)$ is always maximized by $U^{\pi\pi'} = \sigma^{2(L+1)} \delta_{\pi, \pi'}$, which therefore is the solution of Eq. (11) in the GP limit defined by $\alpha \to 0^+$.

**Result 2**. The matrix $U$ obeys the following relation

$$U^{\pi\pi'} = \frac{1}{N} \mathbb{E}[V^{(\text{eff})\pi} \cdot V^{(\text{eff})\pi'\top}] \tag{13}$$

where $V^{(\text{eff})\pi} \in \mathbb{R}^{1 \times N}$ are the effective weights along path $\pi$, defined in Eq. (6).

**Derivation.** See Appendix II.

The derivation, which uses the BPKR technique [29, 33], can be sketched as follows. Computing $\mathbb{E}[f^*]$ under the posterior distribution $p(\Theta)$ involves evaluating a high-dimensional integral in the weights $\Theta$. The idea is to first reduce this computation into an integration over a lower-dimensional, 'macroscopic' variable $U$. Importantly, while $\Theta$ becomes infinite-dimensional as $N \to \infty$, $U$ remains finite-dimensional. The reduced integral is an expectation of the r.h.s. of Eq. (9), treated as a function of $U$, under the distribution $p(U) \propto \exp\left\{-\frac{1}{2} N S(U)\right\}$, where $S$ is the action defined in Eq. (11). Then, this integral can be solved in the thermodynamic limit $N \to \infty$, using the saddle-point method, which implies evaluating Eq. (9) at the $U$ that minimizes the action (cf. Eq. 11). Crucially, the end result is fully characterized by this low-dimensional quantity $U$, commonly called *order parameter* in physics, which has a direct interpretation in terms of the network weights, given by Eq. (13).

In practice, the results obtained in the thermodynamic limit represent a good approximation also for the case of large but finite $N$. In this regard, the scaling of other hyperparameters with $N \to \infty$ is of particular importance, especially the number of training examples $P$. In the GP limit, one considers $P$ finite. This is also called the *infinite-width* thermodynamic limit because in practice, for a given and typically large $P$, it is a good approximation only for very wide networks, when $N \gg P$. In contrast, here we consider the *finite-width* limit in which $P/N = \alpha \in \mathbb{R}^+$ (which includes the GP limit for $\alpha \to 0^+$). As can be seen from Eq. (11), the action gains a new term for $\alpha > 0$, which, as we shall discuss below, is fundamental to account for the learning of an attention paths interplay. Finally, we note that in our numerical experiments (Sec. 4) we will consider Bayesian networks which are overparameterized, i.e. $P < N_0 H^L$, which is the network capacity at fixed query and key weights.

## 3.2 Results interpretation and implications for generalization capability

Eq. (9) is a commonly found expression in thermodynamic theories of Bayesian learning, relating the network's mean predictor to kernel regression. In particular, the theory of kernel regression [39] suggests that generalization improves when the training kernel $K$ is well aligned with the task, meaning its largest principal components (PCs) are well aligned with the vector of training labels $Y$.

Our result for the transformer's kernel (Eq. 10) enables insights into how the transformer architecture favors this kernel-task alignment (Fig. 1(d)). The kernel consists of the sum, weighted by the order parameter $U$, of many *path-path kernels* $C_{\pi\pi'}$, each computing the similarity between the attentioned input on two attention paths $\pi$ and $\pi'$. A notable property of the multi-head architecture is that, despite the number of attention heads growing only linearly with the depth $L$, the number of attention paths grows exponentially $\propto H^L$. Therefore, the network has at its disposal an exponentially large number of path-path kernels, which it can learn, through $U$, to optimally combine into a total kernel with improved task alignment.

This phenomenon, which we term *task-relevant kernel combination*, is indeed predicted by our results Eqs. (11-12). These state that the learned $U$ minimizes a function $S$ (Eq. 11), which, through the energy term $\mathcal{E}$, favors kernel-task alignment. This can be seen by interpreting the energy term (Eq. 12) as the negative log-likelihood of the training labels $Y$ under a centered Gaussian distribution, whose covariance matrix is the training kernel $K$. This negative log-likelihood can be minimized by aligning the largest PCs of the covariance (i.e. the kernel $K$) as much as possible with $Y$ (Fig. 1(c)).

In contrast, in the GP limit $\alpha \to 0^+$, the action $S$ (Eq. 11) consists only of the entropy term $\mathcal{L}$, which does not contain any task relevant information. Its only effect is to attract $U$ towards the GP limit solution $U^{\pi\pi'} = \sigma^{2(L+1)}\delta_{\pi,\pi'}$. Note that, in this limit, the benefits of kernel combination are lost (Fig. 1(d), bottom line): First, out of all the path-path kernels $C_{\pi\pi'}$, only the *same-path kernels* ($\pi = \pi'$) are used, while the *cross-path kernels* ($\pi \neq \pi'$) are discarded; Second, all same-path kernels are weighted equally, without making use of any task-specific information. Note that this is true not only for our simplified model, but also for the full transformer architecture under its known GP limit [25]. A task-relevant kernel combination can therefore only emerge beyond the GP limit, in the finite-width regime $\alpha > 0$ studied in this work.

Finally, our result Eq. (13) relates the order parameter to a macroscopic measure of the network weights, allowing for a direct interpretation of the kernel combination mechanism: correlating the effective weights across paths allows the network to make use of cross-path kernels, while controlling their magnitude allows to weigh the different path-path kernels in a task-relevant manner.

# 4 Experiments

To corroborate our theoretical results, we "train" our model (Eqs. 1-4) by sampling its weights $\Theta$ (i.e. all weights except the fixed query and key weights) from the posterior distribution Eq. (8), using Hamiltonian Monte Carlo sampling (see Appendix F for details). We consider the following two tasks: hidden Markov chain (HMC) classification, and one-shot image classification by in-context learning. The first task is defined on a synthetic dataset. Its purpose is to have a minimal, controllable setting to illustrate the effects of task-relevant kernel combination. In the second task, we will proceed to show analogous effects on classic image datasets (Omniglot [40], MNIST [41], and FashionMNIST [42]), and compare these results with those obtained from the same network trained with standard gradient descent on all of its parameters (i.e. including the query and key weights).

## 4.1 Hidden Markov chain sequence classification

**Task definition.** The HMC classification task is defined as follows (Fig. 2(a)). The $\mu$-th example in the dataset corresponds to an hidden Markov chain $q_1^\mu, \ldots, q_T^\mu$ of length $T = 30$, alternating between two hidden states, $q_t^\mu \in \{+, -\}$. The probability of transition to the opposite state ($\pm \to \mp$) is $p^\mu$. The $\mu$-th chain can belong to one of two classes, labeled $y^\mu = \pm 1$, depending on whether $p^\mu = 0.3$ or $p^\mu = 0.7$, respectively. The input tokens are a noisy, higher dimensional representation of the hidden states. These are given by $x_t^\mu = v_{q_t^\mu} + \eta_t^\mu$, where $v_\pm \in \mathbb{R}^{N_0}$ are two orthogonal feature vectors corresponding to the states "$\pm$", with $\mathcal{O}(1)$ entries, while $\eta_t^\mu$ is a zero-mean Gaussian noise, with $\langle \eta_t^\mu \eta_{t'}^{\mu'\top} \rangle = \delta_{\mu,\mu'}\delta_{t,t'}(\sigma_\parallel^2 P_\parallel^\top \cdot P_\parallel + \sigma_\perp^2 P_\perp^\top \cdot P_\perp)$, where $P_\parallel$ and $P_\perp$ are the projectors along the

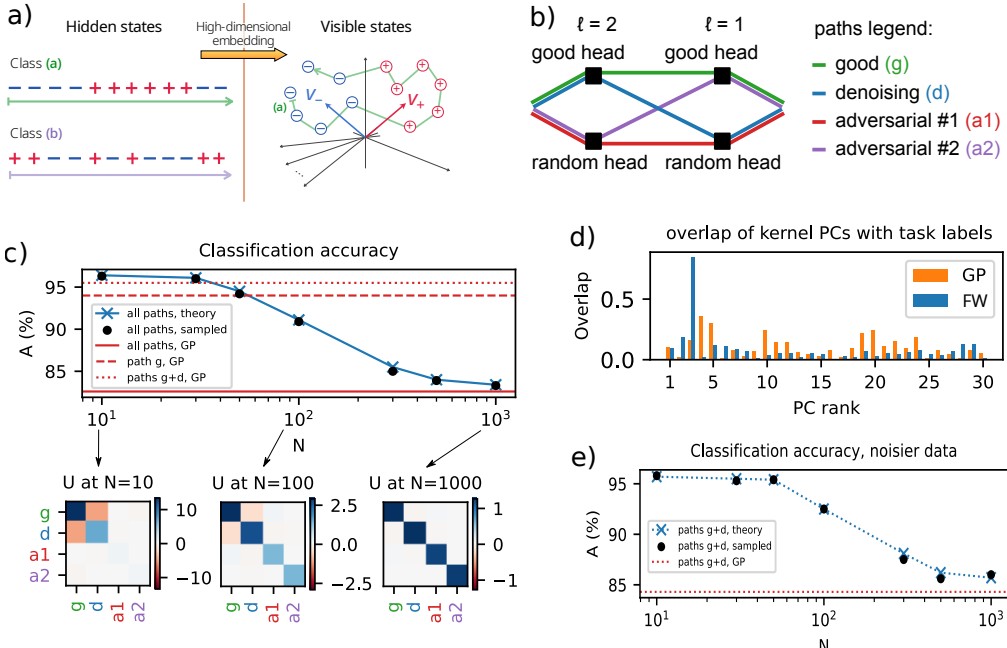

Figure 2: **Hidden Markov chain task. (a)** Illustration of the task. **(b)** Schematics of the network and its attention paths. **(c) Top:** Classification accuracy for varying $N$ (theory: blue crosses, joined by blue line; samples: black dots). Red lines: GP limit for a network consisting of all paths (solid), the good path (dashed), and the good and denoising paths (dotted). **Bottom:** Matrix elements of $U$, for varying $N$. The matrix indices are labeled with the corresponding path name, according to the legend in (b). **(d)** Normalized overlap, or cosine similarity, between the PCs of the kernel $K$ and the vector of task labels $Y$ ($N = 10$: blue; GP limit: orange). PCs are ranked by their eigenvalues, from largest to smallest. Only the first 30 PCs are shown. **(e)** Same as (c), but for increased $\sigma_\perp = 5$ and a network consisting of only the good and denoising paths.

subspace parallel or perpendicular to the plane spanned by $v_+$ and $v_-$. Unless specified, $\sigma_\parallel = \sigma_\perp = 1$. The separate parameterization of the parallel ($\sigma_\parallel$) and perpendicular ($\sigma_\perp$) noise strengths is motivated by their distinct effect on task performance: while the first corrupts information about the underlying hidden states, inevitably putting an upper bound on the classification accuracy, the second can always be filtered out by learning appropriate weights. We use $P = 100$ examples for training. We test the network performance in terms of the classification accuracy $A = \frac{1}{P^*} \sum_\mu \delta_{y^\mu, \text{sign}(\langle f^\mu \rangle)}$, where the sum is over a number $P^* = 1000$ of test examples. Additional task details are given in Appendix G.1.

### 4.1.1 Results

We consider a network of $L = 2$ layers and $H = 2$ heads per layer, with readout from the first token. The network has a total of 4 attention paths, schematically depicted in Fig. 2(b). For this synthetic task, we design the fixed query and key weights, and therefore the network's attention paths, to clearly illustrate the effects of task-relevant kernel combination (for details, see Appendix G.2).

We design the first head of each layer to give rise to a "good" attention path (green path) such that a network consisting of this good path alone achieves a high classification accuracy, $A \sim 94\%$. Along this path, the first head makes use of the Markov nature of the task by attending exclusively to nearby tokens, and only if they correspond to the same hidden state $\pm$; the second head performs uniform attention, effectively counting how many times the first head detected the same-state transition $\pm \to \pm$. In contrast, each layer's second head is initialized randomly. This results in the three remaining paths having chance-level classification accuracy $A \sim 50\%$, when considered in isolation. However, these paths have very different effects, when combined with the good path. We term two of these paths "adversarial" (red and purple paths) because they deteriorate the network performance, while we term the remaining path "denoising" (blue path) because it can be effectively combined with the good path to improve robustness to noisy data.

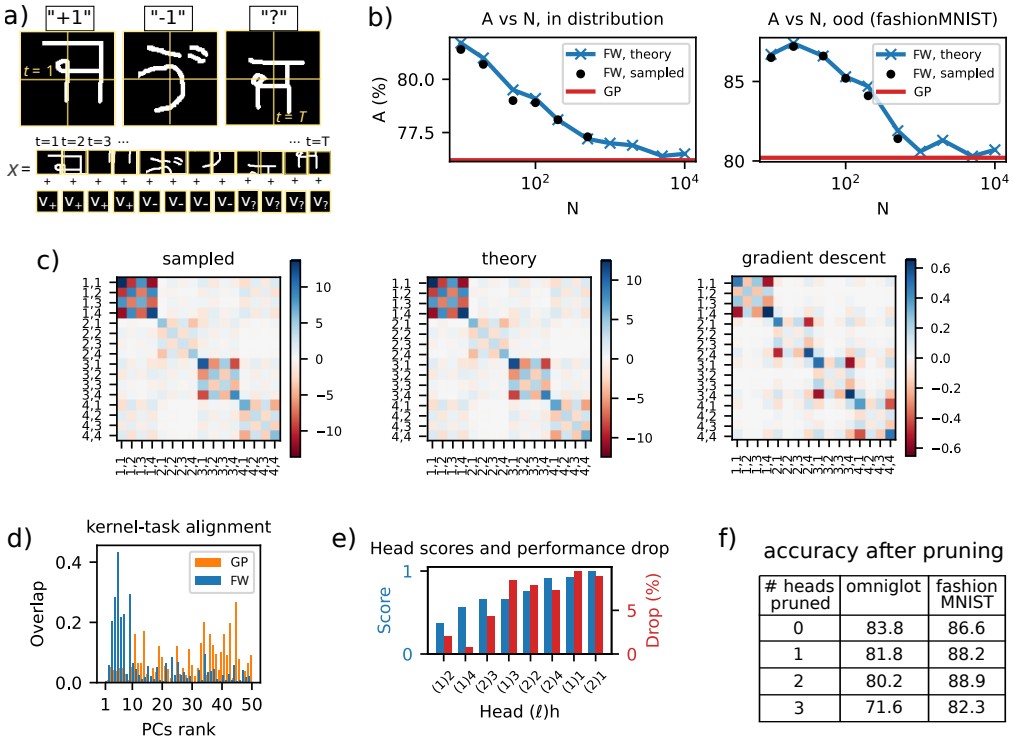

Figure 3: **One-shot image classification task.** **(a)** Scheme of the task. **(b)** Classification accuracy in the GP limit (red line) and the finite-width regime (FW) for varying $N$ (theory: blue crosses, joined by blue line; samples: black dots). **(c)** Matrix elements of $U$. The "theory" and "sampled" $U$s are for $N = 10$. The matrix indices are labeled with the path index $\pi = (h_1, h_2)$. **(d)** Kernel PCs' overlap with the task, in the GP limit and in the finite-width regime for $N = 10$. Only the first 50 PCs are shown. **(e)** Head score (blue) and performance drop (red) after pruning the head, for the model trained with gradient descent. **(f)** Classification accuracy of the model trained with gradient descent, after pruning a growing number of heads, in order of their head score.

In Fig. 2(c, top) we show the network's classification accuracy as a function of the width $N$ (blue, solid curve), compared to the GP limit (red, solid line). At lower $N$, well into the finite-width regime, we observe a considerable improvement in performance with respect to the GP limit. This can be understood in terms of an improved kernel-task alignment, as shown in Fig. 2(d).

This improved alignment is ensured by the order parameter $U$, plotted in Fig. 2(c, bottom) for varying $N$. For $N = 10$, well into the finite-width regime, the order parameter clearly implements the two main benefits of kernel combination: the possibility to weigh the path-path kernels differently, and the ability to make use of the cross-path kernels. The first benefit is particularly apparent in the suppression of all kernels associated with the adversarial paths. In contrast, when $N = 1000$ and the order parameter is very close to its GP limit $U^{\pi\pi'} = \delta_{\pi,\pi'}$, these paths are not suppressed, causing a deterioration in performance compared to that of the good path alone (red, dashed line in Fig. 2(c, top)). The second benefit is apparent in the strong off-diagonals of $U$, anti-correlating the good and denoising paths. We can see that, while also in the GP limit the denoising and good paths combined (dotted, red line in Fig. 2(c, top)) have a better performance than the good path alone (dashed, red line), the performance boost is even higher in the renormalized regime, which makes use of the cross-path kernels. This additional improvement in performance becomes more apparent with noisier data. This is shown in Fig. 2(e), where we plot the classification accuracy of the network consisting of only the good and denoising paths, on data with stronger perpendicular noise $\sigma_\perp = 5$.

## 4.2 One-shot image classification

**Task definition.** The one-shot image classification task (Fig. 3(a)) is formulated in an in-context learning setting. The network is presented with a sequence of three image-label pairs. The first

two images belong to two distinct classes of a categorized dataset (Omniglot, FashionMNIST or MNIST in our case). They are assigned the label "+" or "−" in no particular order. The third image is assigned the label "?", and belongs to one of the classes of the first two images. The network has to output $\pm 1$ according to the label of the matching image. The sequence is fed to the network as follows. Following the idea of the vision transformer (ViT) [8], each image is divided into $p$ patches. The patch $i \in \{1, \ldots, p\}$ of image $a \in \{1, 2, 3\}$ corresponds to the token $x_{(a-1)p+i}$, for a total of $T = 3p$ tokens. We encode the labels $+, -, ?$ using three fixed random vectors $v_+, v_-, v_? \in \mathbb{R}^{N_0}$, which we directly add to each patch (i.e., token) of the corresponding image. We also encode the token position with additive sinusoidal positional encoding [1]. The network is trained on the Omniglot dataset [40], while we test its classification accuracy on both in-distribution (ID) unseen classes of Omniglot, and out-of-distribution (OOD) FashionMNIST dataset (we also report results on MNIST in Appendix H).

### 4.2.1 Results

We consider a network of $L = 2$ attention layers and $H = 4$ heads per layer, with average pooling readout, trained on a subset of $P = 600$ examples from Omniglot (analogous results for a deeper network with $L = 3$, $H = 3$ are also reported in Appendix H.2.2). For the fixed query and key weights required by our Bayesian network, we use the query and key weights obtained from training the same network using gradient descent, with $N = 512$, $G = 128$, and $P = 528k$ (i.e., the entire training set from Omniglot). We refer to Appendix H.1 for further details on this process.

The plots shown in Fig. 3 are analogous to those for the HMC task (Fig. 2), and illustrate analogous kernel combination phenomena. Fig. 3(b) shows the classification accuracy for varying $N$. Again, we observe a performance gap between the finite-width and GP regimes. Interestingly, this improvement in performance is preserved also OOD, on FashionMNIST. Again, Fig. 3(d) shows that the performance gap can be understood in terms of an improved kernel-task alignment: PCs that are well aligned with $Y$ are of higher rank, and have a larger overlap than in the GP limit.

The order parameter (Fig. 3(c), "theory" and "sampled") for $N = 10$ is clearly far from its GP limit, accounting for the improvement in performance observed in the finite-width regime. We observe similar kernel combination phenomena as in the HMC task, with strong off-diagonal elements, and a stronger weighting of certain paths w.r.t. others. Interestingly, the block diagonal structure of the order parameter allows for a simple interpretation of the interplay between paths: correlations mostly occur between paths sharing the same head $h_1$ in the first layer, which also determines which paths are overall enhanced ($h_1 = 1, 3$) or suppressed ($h_1 = 2, 4$).

This structure of the order parameter transfers qualitatively well also to the network trained with gradient descent. In Fig. 3(c, "gradient descent") we show an empirical order parameter, obtained by computing Eq. 13 using a single realization of the network's weights trained with gradient descent. Both the order parameter's block structure and matrix element signs are qualitatively preserved in this empirical estimate. We emphasize that the network is trained with the full set of training examples ($P = 528k$) rather than the restricted one used for the Bayesian network ($P = 600$), and on all learnable parameters including the query and key weights, making this qualitative agreement more relevant to potential applications. One example application is provided below.

**Application: model reduction via head pruning.** Our theory allows us to prune certain heads in the model trained with gradient descent (leading to a model size and compute reduction), with marginal performance loss. This is achieved by using the order parameter to assign a score to each attention head, according to its contribution to kernel combination. The lowest-scoring heads are then pruned from the model. The head $h$ at layer $\ell$ is assigned the score $s^{(\ell)h} = \sum_{\pi, \pi' \in \Pi^{(\ell)h}} |U_{\pi, \pi'}|$, where $\Pi^{(\ell)h}$ is the set of all paths passing through that head, and $U$ is the order parameter derived from theory. Fig. 3(e) shows the score of each head, normalized by the largest one, compared against the drop in classification accuracy caused by pruning that head. Note that the network is not retrained after pruning, but only reevaluated on the test examples. We observe a performance drop qualitatively in line with the head scores. Most importantly, the two lowest scoring heads only cause a marginal drop in performance. In Fig. 3(f) we report the classification accuracy after pruning an increasing number of heads, in order of their score. Up until the first two heads (amounting to $25\%$ of the total number of network parameters), the in-distribution classification accuracy is only marginally worsen. Interestingly, the OOD classification accuracy is even improved, possibly indicating an overspecialization of the pruned heads in solving only the in-distribution task. In Appendix H.2.3 we achieve an analogous size reduction of $25\%$ on a larger model with $H = 8$ heads.

**Agreement with sampled statistics**. Finally, we note that both figures 2(c,e) and 3(b,c) show good agreement of our theory with the mean predictor and order parameter sampled from Eq. 8. While our theory becomes exact in the $N, P \to \infty$ limit, the agreement holds even for small $N = 10$. In particular, in figures 3(b,c), it holds even if $N < H^L \left( H^L + 1 \right) / 2 = 136$, the number of independent entries in the order parameter, which is supposed to be a finite quantity in our theory.

## 5   Conclusion and Discussion

**Conclusion.** We introduce a transformer-like model featuring deep multi-head self-attention, amenable to theoretical characterization within the Bayesian framework. Our results unveil the important role of attention paths in accounting for transformers' remarkable performance. We demonstrate that, in scenarios involving the interplay of attention paths at finite widths, generalization consistently improves compared to the GP regime, where such interplay is absent. Our theory explains this paths interplay in terms of a task-relevant kernel combination mechanism, where the network's total kernel results from the sum of many kernels, specific to pairs of paths, and optimally weighted to improve generalization. This mechanism is confirmed by experiments on both synthetic and real-world sequence classification tasks. More broadly, our results are relevant to the theory of deep learning, as they provide an example of non-scalar kernel renormalization [33, 43] in a widely adopted architecture such as the transformer, illustrating its importance in accounting for network performance. Non-scalar, as opposed to scalar [29], renormalization can affect the network's mean predictor and therefore lead to improved generalization. Our work provides a novel interpretation of its benefits in terms of an optimized kernel-task alignment.

**Interpretability of our Theory.** We provide interpretability to the kernel combination mechanism, by relating it to observable structures in the network weights, specifically to their magnitude and correlations. These predicted structures transfer well outside the Bayesian framework, to the weights of networks trained with gradient descent, broadening the applicability of our theory. As an example, we show that a trained network can be reduced in size with minimal performance loss, by pruning those heads that are deemed less relevant by our theory. The large size reduction achieved (25%) appears in line with observations that a few specialized heads are responsible for most of the network performance [44], and the proposal of head pruning schemes during training [45]. Our theoretical insights may therefore be relevant to the quest for minimalistic and resource-efficient models [46, 47].

**Limitations and Outlook.** The above results are enabled by our theory's ability to go beyond the GP limit [25], as well as incorporating a multi-head, multi-layer architecture—a prerequisite for the very existence of attention paths. However, various limitations could still be addressed, opening for exciting research directions. For example, attention in our model is only a function of the bare input, rather than the previous layer's postactivation, as in standard transformers. In this case, the theoretical challenge would be to disentangle the learning of attention paths interplay from the learning of the attention paths themselves, since now also the attention matrix would depend on the value weights. Another limitation of our model is its linearity in the value weights. It may be possible to heuristically extend the theory to include nonlinear MLP blocks in between attention layers, by replacing the GP path-path kernels appearing in Eq. (10) with the corresponding GP kernels for the nonlinear case—an approach which has proven successful in deep ReLU networks for certain regimes [29]. Introducing nonlinearities, strong feature learning may also emerge [48–50]. Note that, instead, our theory is readily extendable to the case of *linear* MLP blocks, as well as multiple outputs, following [29, 33]. Here we chose a minimal setting focusing only on those renormalization phenomena specific to the transformer architecture. Indeed, the presence of multiple outputs causes the same kind of renormalization independently of the network architecture (i.e., adding two new output indices to the order parameter), while deeper linear blocks would not alter the essence of attention paths interplay, only affecting details in the entropy part of the action. Extending the theory to include skip connections also seems viable. A very open challenge, instead, is to characterize the learning of the query and key weights, which relates to the more general challenge of extending the BPKR technique to nonlinear deep networks. Finally, our approach characterizes the inductive bias imposed by the network architecture on the parameter configurations that fit the training data, but not the bias imposed by a learning algorithm. It would therefore be interesting to import our theory to methods characterizing deep neural networks' training dynamics [51–53].

## Acknowledgements

We acknowledge support of the Swartz Foundation, the Kempner Institute for the Study of Natural and Artificial Intelligence at Harvard University, the Office of Naval Research (ONR) grant No. N0014-23-1-2051, and the Gatsby Charitable Foundation. This research was supported in part by grant NSF PHY-2309135 to the Kavli Institute for Theoretical Physics (KITP). FM was supported by the Simons Foundation (Award Number: 1141576). We have benefitted from helpful discussions with Alexander van Meegen, Haozhe Shan and Qianyi Li.

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

# Appendices

# Part I

# Further discussion on related theory works

The theoretical properties of attention-based models have been investigated from various perspectives in recent years. Different lines of works have studied the expressivity [10, 22, 21, 54], the inductive bias [11, 12] and the training dynamics [17, 16, 18] of attention layers. In particular, [54] shows that multi-layer attention networks are universal approximators for certain classes of functions, such as equivariant sequence-to-sequence functions, while [21] highlights their computational limitations, demonstrating that self-attention cannot model periodic finite-state languages, nor hierarchical structure, unless the number of layers or heads increases with input length. [16, 22] derive error bounds for non-linear models, with trainable queries and keys. As in our work, [19, 10, 55] focus on the training of value matrices. In [19, 10] query and key matrices are fixed and untrainable, while [55] sets them equal to the identity. Several theoretical studies have focused on in-context learning [56–60]. However, the works mentioned above derive generalization bounds and do not provide a tight characterization of the learning curves. A first tight analysis was done in [14], considering a single-layer model with factored self-attention. The authors of [13] provide a tight analysis of a single-layer attention model with trainable tied queries and keys and value weights fixed to the identity. Previous theoretical works have mainly focused on a single transformer block, where attention paths are not defined. For instance, [44] finds that one transformer block can learn different linguistic tasks according to the position of the self-attention layer. [58] shows that a transformer block can learn to encode topical models. [55] shows that one transformer block can encode patch associations. A different line of works has studied attention layers in the infinite-width limit [25, 26]. In particular, [25] establishes an equivalence between Gaussian processes and infinitely-wide multi-layer attention models with infinitely many attention heads. [26] leverages this framework and studies the inductive bias of infinitely-wide transformers towards permutation symmetric functions in sequence space.

# Part II
# Theory

## A  Definitions

We recall the definitions for our transformer model and theory, Sec. 2 and 3 of the main text.

### A.1  Variables and parameters

The hyperparameters are

$$
\begin{aligned}
T &: \quad \text{number of tokens} \\
L &: \quad \text{number of attention layers} \\
H &: \quad \text{number of attention heads per layer} \\
N_0 &: \quad \text{input width} \\
N &: \quad \text{hidden layers width} \\
G &: \quad \text{query-key internal dimension} \\
P &: \quad \text{number of training examples} \\
\mathcal{T} &: \quad \text{Gibbs temperature (theory only)} \\
\sigma &: \quad \text{Gaussian prior variance (theory only)}
\end{aligned}
$$

The network weights are

$$
\begin{aligned}
V^{(0)} \in \mathbb{R}^{N \times N_0} &: \quad \text{input projection weights} \\
V^{(\ell)h} \in \mathbb{R}^{N \times N} &: \quad \text{value weights} \\
a \in \mathbb{R}^{1 \times N} &: \quad \text{readout weights} \\
W_Q^{(\ell)h}, W_K^{(\ell)h} \in \mathbb{R}^{G \times N} &: \quad \text{query and key weights (fixed in the theory)}
\end{aligned}
$$

where $h = 1, \ldots, H$, and $\ell = 1, \ldots, L$.

The network input is

$$
\begin{aligned}
x \in \mathbb{R}^{N_0 \times T} &: \quad \text{input sequence} \\
x_t \in \mathbb{R}^{N_0} &: \quad \text{single token in the sequence}
\end{aligned}
$$

where $t = 1, \ldots, T$.

We append an additional index $\mu = 1, \ldots, P$ to quantities that refer to a specific example in the training set, such as, e.g.

$$
\begin{aligned}
x^\mu \in \mathbb{R}^{N_0 \times T} &: \quad \mu\text{-th example in the training set} \\
y^\mu \in \mathbb{R} &: \quad \mu\text{-th training label}
\end{aligned}
$$

We further define

$$
\begin{aligned}
X &:= \{x^\mu\}_{\mu=1}^P \\
Y &:= \{y^\mu\}_{\mu=1}^P \quad \text{the vector of training labels}
\end{aligned}
$$

### A.2  Model definition

The network performs the following sequence of input transformations (Fig. 4). The input projection layer is defined as

$$
x_t^{(1)} = \frac{1}{\sqrt{N_0}} V^{(0)} \cdot x_t \tag{14}
$$

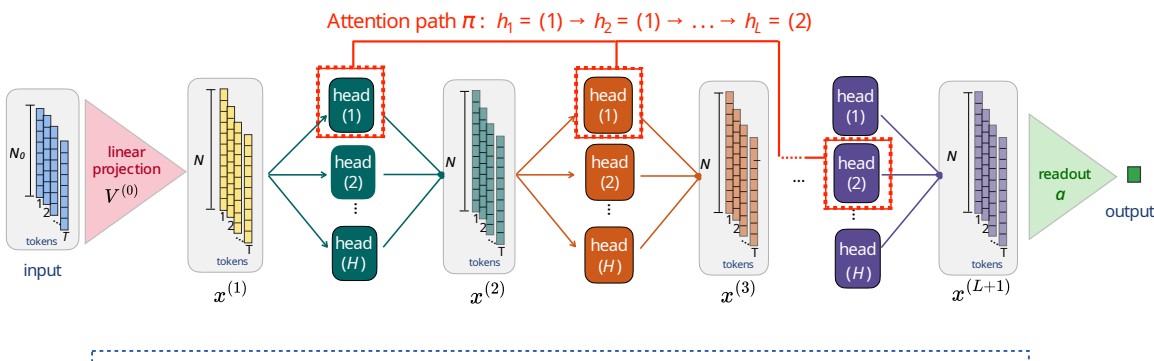

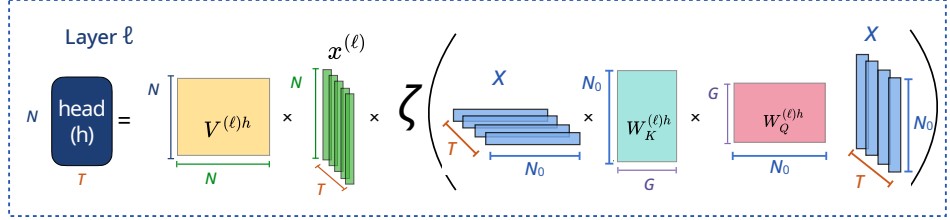

Figure 4: Schematic representation of the architecture under consideration.

The output from each attention layer $\ell = 1, \ldots, L$ is defined recursively as:

$$x_t^{(\ell+1)} = \frac{1}{\sqrt{NH}} \sum_{h=1}^{H} \sum_{s=1}^{T} V^{(\ell)h} \cdot x_s^{(\ell)} \Omega_{st}^{(\ell)h}, \qquad t = 1, \ldots, T \tag{15}$$

where, for each head $h$, we define the attention matrix $\Omega^{(\ell)h} \in \mathbb{R}^{T \times T}$ with matrix elements

$$\Omega_{st}^{(\ell)h} = \zeta \left( \frac{1}{N_0 \sqrt{G}} x_s^{\top} \cdot W_K^{(\ell)h\top} \cdot W_Q^{(\ell)h} \cdot x_t \right) \tag{16}$$

The function $\zeta$ denotes the softmax applied along the direction of the token indexed by $s$. The linear readout is defined as

$$f = \frac{1}{\sqrt{N}} a^{\top} \cdot x_{t^*}^{(L+1)}. \tag{17}$$

where we adopt the unified notation

$$x_{t^*}^{(L+1)} = \begin{cases} \frac{1}{T} \sum_{t=1}^{T} x_t^{(L+1)} & \text{average over all the tokens} \\ x_{t^*}^{(L+1)} & \text{readout from a single token at position } t = t^* \end{cases} \tag{18}$$

in order to consider different options of token readout, adopted in different tasks.

### A.3 Posterior distribution

We are interested in computing the posterior distribution over the network's weights $\Theta := \left( V^{(0)}, \{V^{(\ell)h}\}_{\ell,h=1}^{L,H}, a \right)$. This is analogous to the Gibbs distribution in statistical physics and is defined as

$$p(\Theta) = \frac{1}{Z} \exp \left\{ -\frac{1}{2\mathcal{T}} \sum_{\mu=1}^{P} [f(x^{\mu}, \Theta) - y^{\mu}]^2 - \frac{1}{2\sigma^2} \|\Theta\|^2 \right\}, \tag{19}$$

where $\|\cdot\|$ is the Frobenius norm, and $Z$ indicates the normalization, also called partition function.

### A.4 Attention paths notation

In the main text, we introduce the concept of *attention paths*, by defining a path index

$$\pi := (h_1, \ldots, h_L) \qquad \text{with} \qquad h_1, \ldots, h_L \in \{1, \ldots, H\} \tag{20}$$

which uniquely identifies each possible combination of the head indices, i.e. each possible path through the attention heads (Fig. 4). The network output can be written as

$$f = \frac{1}{\sqrt{H^L N N_0}} \sum_{\pi \in \Pi} V^{(\text{eff})\pi} \cdot V^{(0)} \cdot \xi^{\pi} \tag{21}$$

where $\Pi$ is the set of all possible paths, and we define the *effective weights* as

$$V^{(\text{eff})\pi} := \frac{1}{\sqrt{N^L}} a \cdot V^{(L)h_L} \cdot V^{(L-1)h_{L-1}} \cdot \ldots \cdot V^{(2)h_2} \cdot V^{(1)h_1}, \qquad V^{(\text{eff})\pi} \in \mathbb{R}^{1 \times N} \tag{22}$$

and the *attentioned input*

$$\xi^{\pi} = \sum_{t_0,\ldots,t_{L-1}=1}^{T} x_{t_0} \Omega_{t_0 t_1}^{(1)h_1} \Omega_{t_1 t_2}^{(2)h_2} \cdots \Omega_{t_{L-2}t_{L-1}}^{(L-1)h_{L-1}} \Omega_{t_{L-1}t^*}^{(L)h_L} . \tag{23}$$

In order to present our theoretical results and their derivation, it is useful to also introduce the following notation.

**Partial-path indices.** We indicate with

$$\pi_\ell := (h_\ell, h_{\ell+1}, \ldots, h_L) \qquad \text{with} \qquad h_\ell, \ldots, h_L \in \{1, \ldots, H\} \tag{24}$$

a collection of head indices, from layer $\ell$ up to layer $L$. For $\ell = 1$, $\pi_1 \equiv \pi$ is the path index defined above, Eq. 20, and in the main text. For $\ell \neq 1$, $\pi_\ell$ corresponds more generally to a *partial path*, starting from layer $\ell$. We call the set of such paths $\Pi_\ell$. For $\ell = 1$, $\Pi_1 \equiv \Pi$ is the collection of all complete paths defined above and in the main text.

**Attentioned input at layer $\ell$.** We define the *attentioned input at layer $\ell$*, for $\ell = 1, \ldots, L$, as

$$\xi^{(\ell)\pi_\ell} = \xi^{(\ell)h_\ell h_{\ell+1}\ldots h_L} := \sum_{t_\ell, t_{\ell+1},\ldots,t_L=1}^{T} x_{t_\ell}^{(\ell)} \Omega_{t_\ell t_{\ell+1}}^{(\ell)h_\ell} \cdots \Omega_{t_L, t^*}^{(L)h_L}, \qquad \xi^{(\ell)\pi_\ell} \in \mathbb{R}^N \tag{25}$$

were we remind the reader of the compact notation $t^*$ defined in Eq. 18. The attentioned inputs at subsequent layers are related by

$$\xi^{(\ell)\pi_\ell} = \frac{1}{\sqrt{NH}} \sum_{h_{\ell-1}=1}^{H} V^{(\ell-1)h_{\ell-1}} \cdot \xi^{(\ell-1)h_{\ell-1}\pi_\ell} \tag{26}$$

Recall that $\xi^{(\ell-1)h_{\ell-1}\pi_\ell} := \xi^{(\ell-1)h_{\ell-1}h_\ell h_{\ell+1}\ldots h_L}$, according to the notation defined in Eq. 24.

We also give specific definitions for the cases $\ell = 0$ and $\ell = L+1$. For $\ell = 0$, we define $\xi^{(0)\pi_1} \equiv \xi^{\pi}$, corresponding to the attentioned input defined above, Eq. 23, and in the main text. The following relation holds

$$\xi^{(0)\pi_1} = \frac{1}{\sqrt{N_0}} V^{(0)} \cdot \xi^{(1)\pi_1} \qquad \xi^{(0)\pi_1} \in \mathbb{R}^{N_0} . \tag{27}$$

For $\ell = L+1$ we define

$$\xi^{(L+1)} = \frac{1}{\sqrt{NH}} \sum_{h_L=1}^{H} V^{(L)h_L} \cdot \xi^{(L)h_L} \tag{28}$$

## B    Results enunciation

Here we enunciate our theoretical results, for which a derivation is given in Appendix C and Appendix D.

## B.1 Predictor statistics

We derive (Appendix C) the statistics of the network prediction $f(x^*) := f^*$ on a new test example $x^*$. These are given by

$$\langle f^* \rangle = k^\top \cdot (K + \mathcal{T}\mathbb{1})^{-1} \cdot Y \tag{29}$$

$$\left\langle (\delta f^*)^2 \right\rangle = K_{\text{test}} - k^\top \cdot (K + \mathcal{T}\mathbb{1})^{-1} \cdot k \tag{30}$$

where $\langle \cdot \rangle$ denotes the expectation under the posterior distribution Eq. 19, and $\delta f^* = f^* - \langle f^* \rangle$. The quantities $K_{\text{test}} \in \mathbb{R}$, $k \in \mathbb{R}^P$, and $K \in \mathbb{R}^{P \times P}$ are defined in terms of a kernel function $\mathcal{K} : \mathbb{R}^{N_0 \times T} \times \mathbb{R}^{N_0 \times T} \to \mathbb{R}$. For $\mu, \nu = 1, \ldots, P$ we define $K_{\text{test}} := \mathcal{K}(x^*, x^*)$, $k^\mu := \mathcal{K}(x^*, x^\mu)$, and $K^{\mu\nu} := \mathcal{K}(x^\mu, x^\nu)$.

We derive the following form for the kernel

$$\mathcal{K}(x, x') = \frac{1}{H^L} \sum_{\pi, \pi' \in \Pi} U^{\pi\pi'} C_{\pi\pi'}(x, x') \qquad \text{with} \quad C_{\pi\pi'}(x, x') := \frac{1}{N_0} \xi^{\pi\top}(x) \cdot \xi^{\pi'}(x') , \tag{31}$$

where $\xi^\pi(x)$ is the attentioned input corresponding to an input $x$, while $U$, called *order parameter*, is a path-by-path matrix of size $|\Pi| \times |\Pi|$, where $|\Pi| = H^L$ denotes the size of the set $\Pi$. The value of the order parameter is determined *self-consistently* by minimizing a scalar function $S$, called *action* in the physics literature. This is defined as

$$S\left(\left\{U^{(\ell)}\right\}_{\ell=1}^{L+1}; X, Y\right) := -\mathcal{L}\left(U^{(L+1)}\right) - \sum_{\ell=1}^{L} \mathcal{L}\left(U^{(\ell)} \cdot U_{\text{ext}}^{(\ell+1)-1}\right) + \alpha \mathcal{E}\left(U^{(1)}; X, Y\right) \tag{32}$$

where, for any matrix $M$, we define

$$\mathcal{L}(M) = -\sigma^{-2}\text{Tr}(M) + \ln \det(M) \tag{33}$$

and

$$\mathcal{E}\left(U^{(1)}; X, Y\right) = \frac{1}{P} \ln \det \left(K(U^{(1)}; X) + \mathcal{T}\mathbb{1}\right) + \frac{1}{P} Y^\top \cdot \left(K(U^{(1)}; X) + \mathcal{T}\mathbb{1}\right)^{-1} \cdot Y \tag{34}$$

Note that $K(U^{(1)}; X)$ is simply the kernel defined above, through Eq. 31, where we have explicitly written its dependence on $U^{(1)}$ and $X$. Here we have defined a collection of matrix order parameters $U^{(\ell)}$ for $\ell = 1, \ldots, L$, of size $|\Pi_\ell| \times |\Pi_\ell|$, were $|\Pi_\ell| = H^{L+1-\ell}$ denotes the size of the set $\Pi_\ell$. We called $U^{(1)} \equiv U$. The order parameter $U^{(L+1)}$ is instead a scalar. We also defined $U_{\text{ext}}^{(\ell+1)}$ for $\ell = 1, \ldots, L$ as the matrix of size $|\Pi_\ell| \times |\Pi_\ell|$ with elements

$$U_{\text{ext}}^{(\ell+1)\pi_\ell, \pi_\ell'} = U_{\text{ext}}^{(\ell+1)h_\ell \pi_{\ell+1}, h_\ell' \pi_{\ell+1}'} := U^{(\ell+1)\pi_{\ell+1}, \pi_{\ell+1}'} \delta_{h_\ell, h_\ell'}$$

where $\pi_\ell, \pi_\ell' \in \Pi_\ell$ and $\pi_{\ell+1}, \pi_{\ell+1}' \in \Pi_{\ell+1}$ are the partial-path indices defined in Eq. 24, while $h_\ell, h_{\ell'} = 1, \ldots, H$.

The action must be minimized w.r.t. all of the order parameters. Note that in the main text we defined $S$ as a function of $U^{(1)} \equiv U$ alone. By this we mean $S$ evaluated at its minimum for a fixed $U^{(1)}$, so that only the minimization w.r.t. $U^{(1)}$ is left to perform.

The self-consistent solution for the order parameters is obtained numerically by minimizing Eq. 32 with gradient descent methods. Details on the procedure are given in Appendix E.

## B.2 Order parameter interpretation

We derive (Appendix D) the following expression for the order parameter in terms of the network weights

$$U^{\pi\pi'} = \frac{1}{N} \left\langle V^{(\text{eff})\pi} \cdot V^{(\text{eff})\pi'\top} \right\rangle \qquad \pi, \pi' \in \Pi. \tag{35}$$

where $\langle \cdot \rangle$ denotes statistical averaging over the posterior distribution Eq. 19.

# C Derivation of the predictor statistics

Here we provide the derivation of our result for the predictor statistics, reported in Sec. B.1.

## C.1 Partition function

Let us recall the posterior distribution (Eq. 19)

$$p\left(\Theta\right) = \frac{1}{Z} \exp\left\{-\frac{1}{2\mathcal{T}}\sum_{\mu=1}^{P}\left[f\left(x^{\mu},\Theta\right)-y^{\mu}\right]^{2} - \frac{1}{2\sigma^{2}}\left\|\Theta\right\|^{2}\right\} , \tag{36}$$

where $Z$ indicates the normalization, also called partition function.

It is useful to introduce a vector $t \in \mathbb{R}^{P}$ of $P$ auxiliary variables in order to linearize the squared error in Eq. 36. The partition function then reads

$$Z \propto \int \mathcal{D}t\mathcal{D}\Theta \exp\left(\sum_{\mu=1}^{P} \imath t^{\mu}\left(f^{\mu}-y^{\mu}\right)\right) , \tag{37}$$

where $\imath$ is the imaginary unit, $t^{\mu}$ is the $\mu$-th component of $t$, while $f^{\mu}$ is shorthand for $f\left(x^{\mu},\Theta\right)$, and we use the symbol "$\propto$" to indicate that we are neglecting multiplicative constants. We also defined the Gaussian integration measures $\mathcal{D}t = dt \exp\left(-\frac{\mathcal{T}}{2}\left\|t\right\|^{2}\right)$ and $\mathcal{D}\Theta = d\Theta \exp\left(-\frac{1}{2\sigma^{2}}\left\|\Theta\right\|^{2}\right)$.

Calling $f^{*} := f^{P+1}$, we add a *source term* $\propto \imath t^{P+1}f^{P+1}$ to the exponential in the partition function, corresponding to the test example $x^{*} := x^{P+1}$, obtaining

$$Z\left[\imath t^{P+1}\right] \propto \int \mathcal{D}t\mathcal{D}\Theta \exp\left(-\imath \sum_{\mu=1}^{P} t^{\mu}y^{\mu} + \sum_{\mu=1}^{P+1} \imath t^{\mu}f^{\mu}\right) . \tag{38}$$

In what follows we write $Z\left[\imath t^{P+1}\right]$ as $Z$, without writing its argument explicitly. The partition function Eq. 38 allows us to obtain the predictor statistics on the test example by differentiation

$$\left\langle f^{*}\right\rangle := \left\langle f^{P+1}\right\rangle = \frac{d}{d\left(\imath t^{P+1}\right)} \ln\left(Z\right)\Big|_{t^{P+1}=0} \tag{39}$$

$$\left\langle \delta f^{*}\right\rangle := \left\langle \delta f^{P+1}\right\rangle = \frac{d^{2}}{d\left(\imath t^{P+1}\right)^{2}} \ln\left(Z\right)\Big|_{t^{P+1}=0} \tag{40}$$

where $\delta f^{P+1} = f^{P+1} - \left\langle f^{P+1}\right\rangle$, and $\left\langle \cdot \right\rangle$ denotes statistical averaging over the posterior distribution (Eq. 19). In what follows, we will neglect any multiplicative constants in the partition function, that are independent of $\imath t^{P+1}$, since they are irrelevant for the computation of Eq. 39 and Eq. 40.

## C.2 Weights integration

We proceed to compute the partition function (Eq. 38) by integrating all of the network weights $\Theta$, and finally the auxiliary variable $t$. We use the back-propagating kernel renormalization (BPKR) method [29, 33], which consists in the successive integration of the weights, starting from the last layer down to the first layer.

Let us write Eq. 38 as

$$Z \propto \int \mathcal{D}t \exp\left(-\imath \sum_{\mu=1}^{P} t^{\mu}y^{\mu}\right) I\left(t\right) . \tag{41}$$

where we define

$$I\left(t\right) = \int \mathcal{D}\Theta \exp\left(\sum_{\mu=1}^{P+1} \imath t^{\mu}f^{\mu}\right) \tag{42}$$

Below we focus on computing $I(t)$. In what follows, we will write $I\left(t\right)$ as $I$, without writing its argument explicitly.

### C.2.1 Integration over the readout weights

We start by integrating over the readout weights $a$. In Eq. 42, we substitute $f^\mu = a \cdot \xi^{(L+1)\mu}$, obtaining

$$I = \int \mathcal{D}\Theta \exp\left(\imath \sum_{\mu=1}^{P+1} a \cdot \xi^{(L+1)\mu} t^\mu\right) \tag{43}$$

In what follows, it is useful to define $q^{(L+1)} = \sum_{\mu=1}^{P+1} t^\mu \xi^{(L+1)\mu}$ and in general

$$q^{(\ell)\pi_\ell} = \sum_{\mu=1}^{P+1} t^\mu \xi^{(\ell)\pi_\ell,\mu}, \qquad \pi_\ell \in \Pi_\ell \tag{44}$$

We compute the Gaussian integral in the readout weights $a$, obtaining

$$I \propto \int \mathcal{D}\Theta^{(L)} \exp\left(-\frac{1}{2}\frac{\sigma^2}{N} q^{(L+1)\top} \cdot q^{(L+1)}\right) \tag{45}$$

With $\Theta^{(\ell)}$ we indicate the collection of weights $\Theta^{(\ell)} := \left(V^{(0)}, \left\{V^{(\ell')h}\right\}_{\ell',h=1}^{\ell,H}\right)$, i.e. all weights up to layer $\ell$.

### C.2.2 Integration over the weights at layer $L$

Plugging $q^{(L+1)} = \frac{1}{\sqrt{NH}} \sum_{h_L=1}^{H} V^{(L)h_L} \cdot q^{(L)h_L}$ into Eq. 45 we have

$$I \propto \int \mathcal{D}\Theta^{(L)} \exp\left(-\frac{1}{2}\frac{\sigma^2}{N}\frac{1}{NH} \sum_{h_L,h'_L=1}^{H} q^{(L)h_L\top} V^{(L)h_L\top} \cdot V^{(L)h'_L} q^{(L)h'_L}\right) \tag{46}$$

We perform the integral over the value weights $\left\{V^{(L)h}\right\}_{h=1}^{H}$, by noticing that it is given by the product of $N$ identical integrals of the form

$$\int \left[\prod_{i=1}^{N}\prod_{h_L=1}^{H} dV_{1i}^{(L)h_L}\right] \exp\left(-\frac{1}{2\sigma^2} \sum_{i,i'=1}^{N} \sum_{h_L,h'_L=1}^{H} V_{1i}^{(L)h_L} A_{ih_L,i'h'_L} V_{1i'}^{(L)h'_L}\right) \tag{47}$$

where we defined

$$A_{ih_L,i'h'_L} = \delta_{i,i'}\delta_{h_L,h'_L} + \frac{\sigma^2}{N}\frac{\sigma^2}{NH} q_i^{(L)h_L} q_{i'}^{(L)h'_L} \tag{48}$$

with $i,i' = 1,\ldots,N$ and $h_L,h'_L = 1,\ldots,H$. Here $q_i^{(L)h_L}$ indicates the $i$-th element of the vector $q^{(L)h_L} \in \mathbb{R}^N$. We may consider $A$ as an $NH \times NH$ matrix, whose two indices run over the pairs $(i,h_L)$ and $(i',h'_L)$. With this notation, the result of the value weights integration is

$$I \propto \int \mathcal{D}\Theta^{(L-1)} \exp\left(-\frac{1}{2}N \ln\det(A)\right) \tag{49}$$

Using the matrix determinant lemma[2], we find

$$I \propto \int \mathcal{D}\Theta^{(L-1)} \exp\left(-\frac{1}{2}N \ln\left(1 + R^{(L+1)}\right)\right) \tag{50}$$

where we defined the scalar $R^{(L+1)}$

$$R^{(L+1)} = \frac{\sigma^2}{N}\frac{\sigma^2}{NH} \sum_{h_L=1}^{H} q^{(L)h_L\top} \cdot q^{(L)h_L} \tag{51}$$

---

[2] $\det\left(\mathbb{I} + qq^\top\right) = 1 + q^\top \cdot q$, where $q$ is a vector.

We now enforce the identity of Eq. 51 by Fourier representation of the Dirac delta function, introducing the auxiliary scalar variable $U^{(L+1)}$. We obtain

$$I \propto \int dR^{(L+1)} dU^{(L+1)} \mathcal{D}\Theta^{(L-1)} \exp \left\{ + \frac{1}{2} \frac{N}{\sigma^2} U^{(L+1)} R^{(L+1)} - \frac{1}{2} N \ln \left( 1 + R^{(L+1)} \right) \right.$$

$$\left. - \frac{1}{2} \frac{\sigma^2}{NH} U^{(L+1)} \sum_{h_L=1}^{H} q^{(L)h_L \top} \cdot q^{(L)h_L} \right\} \quad (52)$$

In the statistical physics language, $R^{(L+1)}$ and $U^{(L+1)}$ are called order parameters. Since $N \to \infty$, we can solve the integral in $R^{(L+1)}$ with the saddle point method [61, 62]. The value of $R^{(L+1)}$ at the saddle point is

$$R^{(L+1)} = \sigma^2 U^{(L+1)-1} - 1 \quad (53)$$

Therefore, we obtain

$$I \propto \int dU^{(L+1)} \mathcal{D}\Theta^{(L-1)} \exp \left\{ \frac{N}{2} \mathcal{L} \left( U^{(L+1)} \right) \right.$$

$$\left. - \frac{1}{2} \frac{\sigma^2}{NH} \sum_{\pi_L, \pi_L' \in \Pi_L} U_{\text{ext}}^{(L+1)\pi_L, \pi_L'} q^{(L)\pi_L \top} \cdot q^{(L)\pi_L'} \right\} \quad (54)$$

where we defined the "entropy" term

$$\mathcal{L} \left( U^{(L+1)} \right) = -\frac{1}{\sigma^2} U^{(L+1)} + \ln \left( U^{(L+1)} \right) \quad (55)$$

an we introduced the $|\Pi_L| \times |\Pi_L|$ matrix $U_{\text{ext}}^{(L+1)}$, with matrix elements

$$U_{\text{ext}}^{(L+1)\pi_L, \pi_L'} = U_{\text{ext}}^{(L+1)h_L, h_L'} = U^{(L+1)} \delta_{h_L, h_L'} . \quad (56)$$

Here $\pi_L, \pi_L' \in \Pi_L$ are the partial-path indices defined in Eq. 24, which in this case coincide with $h_L, h_{L'} = 1, \ldots, H$. While the definition of $U_{\text{ext}}^{(L+1)}$ may appear superfluous here, it is useful to perform the proof by induction in Sec. C.2.4.

### C.2.3  Integration over the weights at layer $L - 1$

Next, we perform the integration over the weights at layer $L - 1$. The steps are almost identical to those taken in Sec. C.2.2, but will lead to the introduction of a matrix of order parameters. After this layer, we will be able to provide the results for the integration of the weights at subsequent layers by induction.

Plugging $q^{(L)\pi_L} = \frac{1}{\sqrt{NH}} \sum_{h_{L-1}=1}^{H} q^{(L-1)h_{L-1}\pi_L}$ into Eq. 54, we get

$$I \propto \int dU^{(L+1)} \mathcal{D}\Theta^{(L-1)} \exp \left\{ + \frac{N}{2} \mathcal{L} \left( U^{(L+1)} \right) \right.$$

$$- \frac{\sigma^2}{2N^2 H^2} \sum_{h_{L-1}, h_{L-1}'=1}^{H} \sum_{\pi_L, \pi_L' \in \Pi_L} U_{\text{ext}}^{(L+1)\pi_L, \pi_L'} \times$$

$$\left. \times q^{(L-1)h_{L-1}\pi_L \top} \cdot V^{(L-1)h_{L-1} \top} \cdot V^{(L-1)\pi_L'} \cdot q^{(L-1)h_{L-1}'\pi_L'} \right\} \quad (57)$$

Once again we see that the integral in the value weights $\left\{ V^{(L-1)h} \right\}_{h=1}^{H}$ is given by the product of $N$ identical integrals of the form

$$
\int \left[ \prod_{i=1}^{N} \prod_{h_{L-1}=1}^{H} dV_{1i}^{(L-1)h_{L-1}} \right] \times
$$

$$
\times \exp \left( -\frac{1}{2\sigma^2} \sum_{i,i'=1}^{N} \sum_{h_{L-1},h'_{L-1}=1}^{H} V_{1i}^{(L-1)h_{L-1}} A_{ih_{L-1},i'h'_{L-1}} V_{1i'}^{(L-1)h'_{L-1}} \right) \quad (58)
$$

where we defined

$$
A_{ih_{L-1},i'h'_{L-1}} = \delta_{i,i'} \delta_{h_{L-1},h'_{L-1}}
$$
$$
+ \frac{\sigma^2}{N} \frac{\sigma^2}{NH^2} \sum_{\pi_L,\pi'_L \in \Pi_L} q_i^{(L-1)h_{L-1}\pi_L} U_{\text{ext}}^{(L+1)\pi_L,\pi'_L} q_{i'}^{(L-1)h'_{L-1}\pi'_L} \quad (59)
$$

with $i, i' = 1, \ldots, N$ and $h_L, h'_L = 1, \ldots, H$.

Again, we may consider $A$ as an $NH \times NH$ matrix, whose two indices run over the pairs $(i, h_{L-1})$ and $(i', h'_{L-1})$. With this notation, the result of the value weights integration is

$$
I \propto \int dU^{(L+1)} \mathcal{D}\Theta^{(L-2)} \exp \left( \frac{N}{2} \mathcal{L} \left( U^{(L+1)} \right) - \frac{1}{2} N \ln \det (A) \right) \quad (60)
$$

Using the matrix determinant lemma[3], we find

$$
I \propto \int dU^{(L+1)} \mathcal{D}\Theta^{(L-2)} \exp \left( \frac{N}{2} \mathcal{L} \left( U^{(L+1)} \right) - \frac{1}{2} N \ln \det \left( \mathbb{I} + U_{\text{ext}}^{(L+1)} \cdot R^{(L)} \right) \right) \quad (61)
$$

where we introduced the $|\Pi_L| \times |\Pi_L|$ matrix $R^{(L)}$, with matrix elements defined as

$$
R^{(L)\pi_L,\pi'_L} = \frac{\sigma^2}{N} \frac{\sigma^2}{NH^2} \sum_{h_{L-1}=1}^{H} q^{(L-1)h_{L-1}\pi_L \top} \cdot q^{(L-1)h_{L-1}\pi'_L}, \qquad \pi_L, \pi'_L \in \Pi_L \quad (62)
$$

With the same procedure as in Sec. C.2.2, we introduce the order parameter $R^{(L)}$ and its conjugate $U^{(L)}$, also a $|\Pi_L| \times |\Pi_L|$ matrix. We have

$$
I \propto \int dU^{(L+1)} dR^{(L)} dU^{(L)} \mathcal{D}\Theta^{(L-2)} \exp \left\{ \frac{N}{2} \mathcal{L} \left( U^{(L+1)} \right) \right.
$$
$$
+ \frac{1}{2} \frac{N}{\sigma^2} \text{Tr} \left( U^{(L)} R^{(L)} \right) - \frac{1}{2} N \ln \det \left( \mathbb{I} + U_{\text{ext}}^{(L+1)} \cdot R^{(L)} \right)
$$
$$
\left. - \frac{1}{2} \frac{\sigma^2}{NH^2} \sum_{\pi_L,\pi'_L \in \Pi_L} U^{(L)\pi_L,\pi'_L} \sum_{h_{L-1}=1}^{H} q^{(L-1)h_{L-1}\pi_L \top} \cdot q^{(L-1)h_{L-1}\pi'_L} \right\} \quad (63)
$$

Again, we solve the integral in $R^{(L)}$ with the saddle point method. The value of $R^{(L)}$ at the saddle point is

$$
R^{(L)} = \sigma^2 U^{(L)-1} - U_{\text{ext}}^{(L+1)-1} \quad (64)
$$

Therefore we obtain

$$
I \propto \int dU^{(L+1)} dU^{(L)} \mathcal{D}\Theta^{(L-2)} \exp \left\{ \frac{N}{2} \mathcal{L} \left( U^{(L+1)} \right) + \frac{N}{2} \mathcal{L} \left( U^{(L)} \cdot U_{\text{ext}}^{(L+1)-1} \right) \right.
$$
$$
\left. - \frac{1}{2} \frac{\sigma^2}{NH^2} \sum_{\pi_{L-1},\pi'_{L-1} \in \Pi_{L-1}} U_{\text{ext}}^{(L)\pi_{L-1},\pi'_{L-1}} q^{(L-1)\pi_{L-1}\top} \cdot q^{(L-1)\pi'_{L-1}} \right\} \quad (65)
$$

---

[3] $\det \left( \mathbb{I} + Q \cdot U \cdot Q^{\top} \right) = \det \left( \mathbb{I} + U \cdot Q^{\top} \cdot Q \right)$, with $Q$ and $U$ being $m \times n$ and $n \times n$ matrices respectively.

where we give a more general definition of the entropy Eq. 55, such that it can take a matrix argument

$$\mathcal{L}\left(U^{(L)} \cdot U_{\text{ext}}^{(L+1)-1}\right) = -\frac{1}{\sigma^2} \text{Tr}\left(U^{(L)} \cdot U_{\text{ext}}^{(L+1)-1}\right) + \ln \det\left(U^{(L)} \cdot U_{\text{ext}}^{(L+1)-1}\right) \tag{66}$$

and we introduced the $|\Pi_{L-1}| \times |\Pi_{L-1}|$ matrix $U_{\text{ext}}^{(L)}$, with matrix elements

$$U_{\text{ext}}^{(L)\pi_{L-1},\pi'_{L-1}} = U_{\text{ext}}^{(L)h_{L-1}\pi_L,h'_{L-1}\pi'_L} = U^{(L)\pi_L,\pi'_L}\delta_{h_{L-1},h'_{L-1}}. \tag{67}$$

Here $\pi_{L-1},\pi'_{L-1} \in \Pi_{L-1}$ and $\pi_L,\pi'_L \in \Pi_L$ and are the partial-path indices defined in Eq. 24, while $h_{L-1},h'_{L-1} = 1,\dots,H$.

### C.2.4 Integration over the weights at a generic layer $\ell$

We can now compute the integration over the remaining value weights by induction. We claim that, after integration of the weights at layer $\ell$, $I$ will have the form

$$I \propto \int dU^{(L+1)}dU^{(L)}\dots U^{(\ell+1)}\mathcal{D}\Theta^{(\ell-1)} \exp\left\{\frac{N}{2}\mathcal{L}\left(U^{(L+1)}\right)\right.$$

$$+ \frac{N}{2}\mathcal{L}\left(U^{(L)} \cdot U_{\text{ext}}^{(L+1)-1}\right) + \dots + \frac{N}{2}\mathcal{L}\left(U^{(\ell+1)} \cdot U_{\text{ext}}^{(\ell+2)-1}\right)$$

$$\left. - \frac{1}{2}\frac{\sigma^2}{NH^{L+1-\ell}} \sum_{\pi_\ell,\pi'_\ell \in \Pi_\ell} U_{\text{ext}}^{(\ell+1)\pi_\ell,\pi'_\ell} \sum_{h_\ell=1}^{H} q^{(\ell)\pi_\ell\top} \cdot q^{(\ell)\pi'_\ell}\right\} \tag{68}$$

Here we defined a collection of matrix order parameters, one for each integrated layer. The order parameter $U^{(\ell)}$ is a partial-path-by-partial-path matrix of size $|\Pi_\ell| \times |\Pi_\ell|$, were $|\Pi_\ell| = H^{L+1-\ell}$ denotes the size of the set $\Pi_\ell$. We also defined $U_{\text{ext}}^{(\ell+1)}$ as the matrix of size $|\Pi_\ell| \times |\Pi_\ell|$ with elements

$$U_{\text{ext}}^{(\ell+1)\pi_\ell,\pi'_\ell} = U_{\text{ext}}^{(\ell+1)h_\ell\pi_{\ell+1},h'_\ell\pi'_{\ell+1}} := U^{(\ell+1)\pi_{\ell+1},\pi'_{\ell+1}}\delta_{h_\ell,h'_\ell} \tag{69}$$

where $\pi_\ell,\pi'_\ell \in \Pi_\ell$ and $\pi_{\ell+1},\pi'_{\ell+1} \in \Pi_{\ell+1}$ are the partial-path indices defined in Eq. 24, while $h_\ell,h_{\ell'} = 1,\dots,H$.

Eq. 68 is verified for layer $\ell = L - 1$, which we derived in Sec. C.2.3. The induction step, integrating over the weights at layer $\ell - 1$, is done by plugging $q^{(\ell)\pi_\ell} = \sum_{h_{\ell-1}=1}^{H} V^{(\ell-1)h_{\ell-1}} \cdot q^{(\ell-1)h_{\ell-1}\pi_\ell}$ into Eq. 68 and applying exactly the same steps presented in Sec. C.2.3.

### C.3 Integration of the auxiliary variable $t$

After integrating all of the network weights, we have

$$Z \propto \int \mathcal{D}t \exp\left(-i\sum_{\mu=1}^{P} t^\mu y^\mu\right) I(t). \tag{70}$$

with

$$I(t) \propto \int \left[\prod_{\ell=1}^{L+1} dU^{(\ell)}\right] \exp\left\{\frac{N}{2}\mathcal{L}\left(U^{(L+1)}\right) + \frac{N}{2}\sum_{\ell=1}^{L}\mathcal{L}\left(U^{(\ell)} \cdot U_{\text{ext}}^{(\ell+1)-1}\right)\right.$$

$$\left. - \frac{1}{2}\frac{\sigma^2}{NH^L} \sum_{\pi_1,\pi'_1 \in \Pi_1} U^{(1)\pi_1,\pi'_1} q^{(0)\pi_1\top} \cdot q^{(0)\pi'_1}\right\} \tag{71}$$

plugging $q^{(0)\pi_1} = \sum_{\mu=1}^{P+1} \xi^{(0)\pi_1,\mu}t^\mu$ into Eq. 71, we see that we need to perform the following integral in $t \in \mathbb{R}^P$

$$\exp\left(-\frac{1}{2}t^{P+1}K_{\text{test}}t^{P+1}\right)\int\left[\prod_{\mu=1}^{P} dt^\mu\right]\exp\left(-\frac{1}{2}t^\top \cdot (K + \mathcal{T}\mathbb{I}) \cdot t - \left(t^{P+1}k + iY\right)^\top \cdot t\right) \tag{72}$$

where for convenience we report here the kernel definitions given in Sec. B.1. The quantities $K_{\text{test}} \in \mathbb{R}$, $k \in \mathbb{R}^P$, and $K \in \mathbb{R}^{P \times P}$ are defined in terms of a kernel function $\mathcal{K} : \mathbb{R}^{N_0 \times T} \times \mathbb{R}^{N_0 \times T} \to \mathbb{R}$. For $\mu, \nu = 1, \ldots, P$ we define $K_{\text{test}} := \mathcal{K}(x^*, x^*)$, $k^\mu := \mathcal{K}(x^*, x^\mu)$, and $K^{\mu\nu} := \mathcal{K}(x^\mu, x^\nu)$. The form of the kernel is

$$\mathcal{K}(x, x') = \frac{1}{H^L} \sum_{\pi_1, \pi_1' \in \Pi_1} U^{(1)\pi_1, \pi_1'} C_{\pi_1 \pi_1'}(x, x') , \tag{73}$$

with

$$C_{\pi_1 \pi_1'}(x, x') := \frac{1}{N_0} \xi^{(0)\pi_1 \top}(x) \cdot \xi^{(0)\pi_1'}(x') , \tag{74}$$

Computing the Gaussian integral (Eq. 72) we obtain

$$Z \propto \int \left[ \prod_{\ell=1}^{L+1} dU^{(\ell)} \right] \exp\left( -\frac{N}{2} S\left( \left\{ U^{(\ell)} \right\}_{\ell=1}^{L+1} \right) \right)$$
$$\exp\left( \frac{1}{2} \imath t^{P+1} \left[ K_{\text{test}} - k^\top \cdot (K + \mathcal{T}\mathbb{I})^{-1} \cdot k \right] \imath t^{P+1} + \imath t^{P+1} k^T \cdot (K + \mathcal{T}\mathbb{I})^{-1} \cdot Y \right) \tag{75}$$

where we report here for convenience the definition of the action $S$ given in Sec. B.1

$$S\left( \left\{ U^{(\ell)} \right\}_{\ell=1}^{L+1} \right) := -\mathcal{L}\left( U^{(L+1)} \right) - \sum_{\ell=1}^{L} \mathcal{L}\left( U^{(\ell)} \cdot U_{\text{ext}}^{(\ell+1)-1} \right) + \alpha \mathcal{E}\left( U^{(1)} \right) \tag{76}$$

with

$$\mathcal{E}\left( U^{(1)} \right) = \frac{1}{P} \ln \det (K + \mathcal{T}\mathbb{I}) + \frac{1}{P} Y^\top \cdot (K + \mathcal{T}\mathbb{I})^{-1} \cdot Y \tag{77}$$

In the limit $N, P \to \infty$, $\frac{P}{N} \to \alpha \in \mathbb{R}^+$, we solve the integrals in $\left\{ U^{(\ell)} \right\}_{\ell=1}^{L+1}$ with the saddle point method [61, 62]. The partition function therefore takes the final form

$$Z \propto \int \exp\left( \frac{1}{2} \imath t^{P+1} \left[ K_{\text{test}} - k^\top \cdot (K + \mathcal{T}\mathbb{I})^{-1} \cdot k \right] \imath t^{P+1} + \imath t^{P+1} k^\top \cdot (K + \mathcal{T}\mathbb{I})^{-1} \cdot Y \right) \tag{78}$$

where we recall that $K_{\text{test}}$, $k$, and $K$ all depend on $U^{(1)}$, which must be evaluated at the minimum of the action Eq. 76 with respect to all of its arguments $\left\{ U^{(\ell)} \right\}_{\ell=1}^{L+1}$.

Differentiating by $\imath t^{P+1}$ the partition function Eq. 78 (see Eq. 39 and Eq. 40), we obtain the results for the predictor mean (Eq. 29) and variance (Eq. 30) presented in Sec. B.1.

## D  Derivation of the order parameter interpretation

Here we provide the derivation of our result on the order parameter interpretation, exposed in Sec. B.2. The derivation is almost identical to that for the predictor statistics given in Sec. C. What follows below should be considered as a continuation of Sec. C, to which we refer for definitions.

For convenience, we report here the result we want to derive

$$U^{(1)\pi_1 \pi_1'} = \frac{1}{N} \left\langle V^{(\text{eff})\pi_1} \cdot V^{(\text{eff})\pi_1' \top} \right\rangle . \tag{79}$$

where $\pi_1, \pi_1' \in \Pi_1$ are the path indices defined in Eq. 24, while $\langle \cdot \rangle$ denotes statistical averaging over the posterior distribution Eq. 19. We also recall the definition of the network effective weights

$$V^{(\text{eff})\pi_1} := \frac{1}{\sqrt{N^L}} a \cdot V^{(L)h_L} \cdot V^{(L-1)h_{L-1}} \cdot \ldots \cdot V^{(2)h_2} \cdot V^{(1)h_1}, \qquad V^{(\text{eff})\pi_1} \in \mathbb{R}^{1 \times N} \tag{80}$$

### D.1 Partition function

As in Sec. C.1, we start from the partition function

$$Z \propto \int \mathcal{D}t\mathcal{D}\Theta \exp\left(\sum_{\mu=1}^{P} \imath t^{\mu}\left(f^{\mu} - y^{\mu}\right)\right) . \tag{81}$$

To the partition function, we add a *source term* $\propto \sum_{\pi_1 \in \Pi_1} V^{(\text{eff})\pi_1} \cdot q_*^{(1)\pi_1}$, with $q_*^{(1)\pi_1} \in \mathbb{R}^N$

$$Z \propto \int \mathcal{D}t\mathcal{D}\Theta \exp\left(-\imath \sum_{\mu=1}^{P} t^{\mu}y^{\mu} + \sum_{\mu=1}^{P} \imath t^{\mu}f^{\mu} + \frac{\imath}{\sqrt{NH^L}} \sum_{\pi_1 \in \Pi_1} V^{(\text{eff})\pi_1} \cdot q_*^{(1)\pi_1}\right) . \tag{82}$$

such that differentiating by $q_*^{(1)\pi_1}$ allows us to obtain

$$\frac{1}{N}\left\langle V^{(\text{eff})\pi} \cdot V^{(\text{eff})\pi'\top}\right\rangle = -H^L \frac{1}{Z} \sum_{i=1}^{N} \frac{dZ}{dq_{*,i}^{(0)\pi_1} dq_{*,i}^{(0)\pi'_1}}\bigg|_{q_*^{(0)}=0} \tag{83}$$

where $q_{*,i}^{(0)\pi_1}$ indicates the $i$-th component of the vector $q_*^{(0)\pi_1} \in \mathbb{R}^N$, while $q_*^{(0)} \in \mathbb{R}^{N\times H^L}$ is the matrix whose $\pi_1$-th component is $q_*^{(0)\pi_1}$.

As in Sec. C, we proceed to compute the partition function (Eq. 82) by integrating all of the network weights $\Theta$, and finally the auxiliary variable $t$. We make the following observation. In Eq. 82, we can write explicitly $f^{\mu} = \frac{1}{\sqrt{NH^L}}\sum_{\pi_1 \in \Pi_1} V^{(\text{eff})\pi_1} \cdot \xi^{(1)\pi_1,\mu}$. Furthermore, as in Sec. C.2.1 we can define $q^{(L+1)} = \sum_{\mu=1}^{P} t^{\mu}\xi^{(L+1)\mu}$ and in general

$$q^{(\ell)\pi_\ell} = \sum_{\mu=1}^{P} t^{\mu}\xi^{(\ell)\pi_\ell,\mu}, \qquad \pi_\ell \in \Pi_\ell \tag{84}$$

Then Eq. 82 takes the form

$$Z \propto \int \mathcal{D}t\mathcal{D}\Theta \exp\left(-\imath \sum_{\mu=1}^{P} t^{\mu}y^{\mu} + \frac{\imath}{\sqrt{NH^L}} \sum_{\pi_1 \in \Pi_1} V^{(\text{eff})\pi_1} \cdot \left(q^{(1)\pi_1} + q_*^{(1)\pi_1}\right)\right) \tag{85}$$

Renaming $q^{(1)\pi_1} + q_*^{(1)\pi_1} \to q^{(1)\pi_1}$, we see that the steps for computing Eq. 85 are identical to those in Sec. C, until the integration over the input projection weights $V^{(0)}$.

### D.2 Integration of the input projection weights

After integrating all of the network weights except the input projection $V^{(0)}$ we have

$$Z \propto \int \mathcal{D}t \exp\left(-\imath \sum_{\mu=1}^{P} t^{\mu}y^{\mu}\right) I(t) . \tag{86}$$

with

$$I(t) \propto \int \mathcal{D}V^{(0)} \int \left[\prod_{\ell=2}^{L+1} dU^{(\ell)}\right] \exp\left\{+\frac{N}{2}\mathcal{L}\left(U^{(L+1)}\right) + \frac{N}{2}\sum_{\ell=2}^{L}\mathcal{L}\left(U^{(\ell)} \cdot U_{\text{ext}}^{(\ell+1)-1}\right)\right.$$
$$\left. -\frac{1}{2}\frac{\sigma^2}{NH^L}\sum_{\pi_1,\pi'_1 \in \Pi_1} U_{\text{ext}}^{(2)\pi_1,\pi'_1} q^{(1)\pi_1\top} \cdot q^{(1)\pi'_1}\right\} \tag{87}$$

We now substitute $q^{(1)\pi_1} \to q^{(1)\pi_1} + q_*^{(1)\pi_1}$ in Eq. 87, as well as $q^{(1)\pi_1} = \frac{1}{\sqrt{N_0}}\sum V^{(0)} \cdot q^{(0)\pi_1}$ obtaining

$$I\left(t\right) \propto \int \mathcal{D}V^{(0)} \int \left[\prod_{\ell=2}^{L+1} dU^{(\ell)}\right] \exp\left\{ +\frac{N}{2}\mathcal{L}\left(U^{(L+1)}\right) + \frac{N}{2}\sum_{\ell=2}^{L}\mathcal{L}\left(U^{(\ell)}\cdot U_{\text{ext}}^{(\ell+1)-1}\right)\right.$$

$$-\frac{1}{2}\frac{\sigma^2}{NH^L}\sum_{\pi_1,\pi_1'\in\Pi_1} U_{\text{ext}}^{(2)\pi_1,\pi_1'} q_*^{(1)\pi_1\top}\cdot q_*^{(1)\pi_1'}$$

$$-\frac{1}{2}\frac{\sigma^2}{NN_0 H^L}\sum_{\pi_1,\pi_1'\in\Pi_1} U_{\text{ext}}^{(2)\pi_1,\pi_1'} q^{(0)\pi_1\top}\cdot V^{(0)\top}\cdot V^{(0)}\cdot q^{(0)\pi_1'}$$

$$\left. -\frac{\sigma^2}{N\sqrt{N_0}H^L}\sum_{\pi_1,\pi_1'\in\Pi_1} U_{\text{ext}}^{(2)\pi_1,\pi_1'} q_*^{(1)\pi_1\top}\cdot V^{(0)}\cdot q^{(0)\pi_1'}\right\} \quad (88)$$

The integral in $V^{(0)}$ in Eq. 88 has the form

$$\int \left[\prod_{i=1}^{N}\prod_{j=1}^{N_0} dV_{ij}^{(0)}\right] \exp\left(-\frac{1}{2\sigma^2}\sum_{i=1}^{N}\sum_{j,j'=1}^{N_0} V_{ij}^{(0)} A_{j,j'} V_{ij'}^{(0)} - \sum_{i=1}^{N}\sum_{j=1}^{N_0} J_{ij} V_{ij}^{(0)}\right) \quad (89)$$

where we defined

$$A_{j,j'} = \delta_{j,j'} + \frac{\sigma^2}{N}\frac{\sigma^2}{N_0 H^L}\sum_{\pi_1,\pi_1'\in\Pi_1} U_{\text{ext}}^{(2)\pi_1,\pi_1'} q_j^{(0)\pi_1} q_{j'}^{(0)\pi_1'} \quad (90)$$

$$J_{ij} = \frac{\sigma^2}{N\sqrt{N_0}H^L}\sum_{\pi_1,\pi_1'\in\Pi_1} U_{\text{ext}}^{(2)\pi_1,\pi_1'} q_{*,i}^{(1)\pi_1} q_j^{(0)\pi_1'} \quad (91)$$

where $i = 1,\ldots,N$, while $j,j' = 1,\ldots,N_0$, and with the notation $q_{*,i}^{(1)\pi_1}$ and $q_i^{(0)\pi_1}$ we indicate the $i$-th component of the vectors $q_*^{(1)\pi_1}$ and $q^{(0)\pi_1}$ respectively.

We may consider $A_{j,j'}$ as the elements of an $N_0 \times N_0$ matrix $A$. With this notation, the partition function after performing the integral Eq. 89 is

$$Z \propto \int \mathcal{D}t \exp\left(-\imath\sum_{\mu=1}^{P} t^\mu y^\mu\right) \int \left[\prod_{\ell=2}^{L+1} dU^{(\ell)}\right] \times$$

$$\times \exp\left\{\frac{N}{2}\mathcal{L}\left(U^{(L+1)}\right) + \frac{N}{2}\sum_{\ell=2}^{L}\mathcal{L}\left(U^{(\ell)}\cdot U_{\text{ext}}^{(\ell+1)-1}\right) - \frac{N}{2}\ln\det(A)\right.$$

$$\left. -\frac{1}{2}\frac{\sigma^2}{NH^L}\sum_{\pi_1,\pi_1'\in\Pi_1} q_*^{(1)\pi_1'\top} q_*^{(1)\pi_1}\left(U_{\text{ext}}^{(2)\pi_1,\pi_1'} - \frac{\sigma^2}{N}\frac{\sigma^2}{N_0 H^L}\sum_{\rho_1,\rho_1'\in\Pi_1} U_{\text{ext}}^{(2)\pi_1,\rho_1} q_j^{(0)\rho_1} A_{j,j'}^{-1} q_{j'}^{(0)\rho_1'} U_{\text{ext}}^{(2)\rho_1',\pi_1'}\right)\right\}$$

$$(92)$$

We now differentiate the partition function in Eq. 92 by $q_*^{(1)}$, as specified by Eq. 83, obtaining

$$\frac{1}{N}\left\langle V^{(\text{eff})\pi_1}\cdot V^{(\text{eff})\pi_1'\top}\right\rangle = \frac{1}{Z\left(q_*^{(0)}=0\right)}\int \mathcal{D}t \exp\left(-\imath\sum_{\mu=1}^{P} t^\mu y^\mu\right)\int \left[\prod_{\ell=2}^{L+1} dU^{(\ell)}\right] \times$$

$$\sigma^2\left(U_{\text{ext}}^{(2)\pi_1,\pi_1'} - \frac{\sigma^2}{N}\frac{\sigma^2}{N_0 H^L}\sum_{\rho_1,\rho_1'\in\Pi_1} U_{\text{ext}}^{(2)\pi_1,\rho_1} q_j^{(0)\rho_1} A_{j,j'}^{-1} q_{j'}^{(0)\rho_1'} U_{\text{ext}}^{(2)\rho_1',\pi_1'}\right) \times$$

$$\times \exp\left(+\frac{N}{2}\mathcal{L}\left(U^{(L+1)}\right) + \frac{N}{2}\sum_{\ell=2}^{L}\mathcal{L}\left(U^{(\ell)}\cdot U_{\text{ext}}^{(\ell+1)-1}\right) - \frac{N}{2}\ln\det(A)\right) \quad (93)$$

The remaining steps are the same as in Sec. C. We use the matrix determinant lemma[4] and the Woodbury matrix identity[5] respectively to express $\ln \det(A)$ and $A^{-1}$ in terms of a $|\Pi_1| \times |\Pi_1|$ matrix $R^{(1)}$ with elements

$$R^{(1)\pi_1,\pi_1'} = \frac{\sigma^2}{N} \frac{\sigma^2}{N_0 H^L} q^{(0)\pi_1 \top} \cdot q^{(0)\pi_1'} \qquad \pi_1, \pi_1' \in \Pi_L$$

whose identity we enforce by Fourier representation of the Dirac delta function, introducing the auxiliary $|\Pi_1| \times |\Pi_1|$ matrix $U^{(1)}$. The result of these operations is

$$\frac{1}{N} \left\langle V^{(\mathrm{eff})\pi_1} \cdot V^{(\mathrm{eff})\pi_1' \top} \right\rangle = \frac{1}{Z\left(q_*^{(0)} = 0\right)} \int \mathcal{D}t \exp\left(-\imath \sum_{\mu=1}^{P} t^\mu y^\mu\right) \int \left[\prod_{\ell=1}^{L+1} dU^{(\ell)}\right] dR^{(1)} \times$$

$$\sigma^2 \left[U_{\mathrm{ext}}^{(2)} \left(\mathbb{I} - R^{(1)} \cdot U_{\mathrm{ext}}^{(2)} + R^{(1)} \cdot \left(\mathbb{I} + U_{\mathrm{ext}}^{(2)} \cdot R^{(1)}\right)^{-1} \cdot U_{\mathrm{ext}}^{(2)} \cdot R^{(1)} \cdot U_{\mathrm{ext}}^{(2)}\right)\right]^{\pi_1,\pi_1'} \times$$

$$\times \exp\Bigg\{ + \frac{N}{2}\mathcal{L}\left(U^{(L+1)}\right) + \frac{N}{2}\sum_{\ell=2}^{L} \mathcal{L}\left(U^{(\ell)} \cdot U_{\mathrm{ext}}^{(\ell+1)-1}\right)$$

$$+ \frac{1}{2}\frac{N}{\sigma^2}\mathrm{Tr}\left(U^{(1)} R^{(1)}\right) - \frac{1}{2}N\ln\det\left(\mathbb{I} + U_{\mathrm{ext}}^{(2)} \cdot R^{(1)}\right)$$

$$- \frac{1}{2}\frac{\sigma^2}{NH^L}\sum_{\pi_1,\pi_1'\in\Pi_1} U^{(1)\pi_1,\pi_1'} q^{(0)\pi_1\top} \cdot q^{(0)\pi_1'} \Bigg\} \quad (94)$$

As in Sec. C, we solve the integral in $R^{(1)}$ with the saddle point method. The value of $R^{(1)}$ at the saddle point is

$$R^{(1)} = \sigma^2 U^{(1)-1} - U_{\mathrm{ext}}^{(2)-1} \quad (95)$$

Plugging this back into Eq. 94 we obtain

$$\frac{1}{N} \left\langle V^{(\mathrm{eff})\pi_1} \cdot V^{(\mathrm{eff})\pi_1' \top} \right\rangle = \frac{1}{Z\left(q_*^{(0)} = 0\right)} \int \mathcal{D}t \exp\left(-\imath \sum_{\mu=1}^{P} t^\mu y^\mu\right) \int \left[\prod_{\ell=1}^{L+1} dU^{(\ell)}\right] \times$$

$$\times U^{(1)\pi_1,\pi_1'} \exp\Bigg\{ \frac{N}{2}\mathcal{L}\left(U^{(L+1)}\right) + \frac{N}{2}\sum_{\ell=1}^{L} \mathcal{L}\left(U^{(\ell)} \cdot U_{\mathrm{ext}}^{(\ell+1)-1}\right)$$

$$- \frac{1}{2}\frac{\sigma^2}{NH^L}\sum_{\pi_1,\pi_1'\in\Pi_1} U^{(1)\pi_1,\pi_1'} q^{(0)\pi_1\top} \cdot q^{(0)\pi_1'} \Bigg\} \quad (96)$$

### D.3 Integration of the auxiliary variable $t$

The calculation of the integral in $t$ follows that in Sec. C.3. We obtain

$$\frac{1}{N} \left\langle V^{(\mathrm{eff})\pi_1} \cdot V^{(\mathrm{eff})\pi_1' \top} \right\rangle = \left\langle U^{(1)\pi_1,\pi_1'} \right\rangle_{U^{(1)} \sim p\left(U^{(1)}\right)} \quad (97)$$

where $\langle \cdot \rangle_{U^{(1)} \sim p\left(U^{(1)}\right)}$ denotes the expectation under the distribution

$$p\left(U^{(1)}\right) \propto \int \left[\prod_{\ell=2}^{L+1} dU^{(\ell)}\right] \exp\left(-\frac{N}{2}S\left(\left\{U^{(\ell)}\right\}_{\ell=1}^{L+1}\right)\right) \quad (98)$$

---

[4] $\det\left(\mathbb{I} + Q \cdot U \cdot Q^\top\right) = \det\left(\mathbb{I} + U \cdot Q^\top \cdot Q\right)$, with $Q$ and $U$ being $m \times n$ and $n \times n$ matrices respectively.
[5] $\left(\mathbb{I} + Q \cdot U \cdot Q^\top\right)^{-1} = \mathbb{I} - Q \cdot \left(\mathbb{I} + U \cdot Q^\top \cdot Q\right)^{-1} \cdot U \cdot Q^T$, with $Q$ and $U$ being $m \times n$ and $n \times n$ matrices respectively.

where the action $S$ is the same defined in Sec. C.3, Eq. 76. Exactly as in Sec. C.3, we compute the expectation using the saddle point method, under the limit $N, P \to \infty$, $\frac{P}{N} \to \alpha \in \mathbb{R}^+$. We therefore arrive at the final result

$$\frac{1}{N} \left\langle V^{(\text{eff})\pi_1} \cdot V^{(\text{eff})\pi_1'\top} \right\rangle = U^{(1)\pi_1,\pi_1'} \tag{99}$$

where $U^{(1)}$ is the value at the saddle point of $S$.

# Part III

# Experiments

## E  Numerical evaluation of the order parameter

The order parameter in our theory is defined self-consistently as the minimum of an action, Eq. 32. We determine this order parameter numerically, by minimizing the action using the Adam optimizer [63]. As we describe in Appendix H.2.1, for each $N$ (out of 10 choices among $N \in \{a\,10^b, 10^4; a \in \{1, 2, 5\}, b \in \{1, 2, 3\}\}$), we search for the optimal temperature among 10 choices $\mathcal{T} \in \{a\,10^{-b}, 1.0, 1.5; a \in \{1, 2.5, 5, 7.5\}, b \in \{1, 2\}\}$, resulting in 100 total configurations. Each such run takes less than 12 hours on a single A100-40GB GPU. The learning rate is optimized for each configuration by sweeping among $\{10^{-4}, a\,10^{-b}; a \in \{1, 5, 8\}, b \in \{0, 1, 2, 3\}\}$ for 10 iterations at the beginning of each run, and by selecting the one that achieves the lowest energy term (averaged over these 10 first optimization steps).

## F  Hamiltonian Monte Carlo sampling

We sample the network weights from the posterior distribution (Eq. 19) using Hamiltonian Monte Carlo sampling [64]. Specifically, we use the NumPyro implementation of the No U-Turn Sampler (NUTS) [65, 66].

For $N \leq 100$, we run 10 independent chains, consisting of 1000 warm-up steps and 1000 sampling steps. We keep one sample every 10, for a total of 1000 samples. Due to limited computational resources, the number of samples is smaller and varies for $N > 100$, while we did not take samples at all for very large $N \geq 1000$. Note however, that the large $N$ regime is the least relevant to sample, since the network is approaching the GP limit. The most important validation of our theory is performed for the smaller values of $N$.

Each run sampling a model at a given $N$ takes less than 12 hours on a single A100-40GB GPU. The number of runs to sample all model widths for the hidden Markov chain classification task and the one-shot image classification task is 12.

Regarding the temperature $\mathcal{T}$ of the Bayesian posterior Eq. 19, this is set differently depending on the task. For the HMC task, we find the temperature $\mathcal{T}$ to not be relevant for improving the network's classification accuracy. We therefore set it to a small, but finite value of $\mathcal{T} = 0.01$. In contrast, for the one-shot image classification task, we find tuning the temperature to be particularly important to prevent overfitting. Therefore, we always tune the temperature to the value giving the optimal classification accuracy for the given network depth $N$. We refer to Appendix H.2.1 for details on the temperature values and its optimization process.

## G  Hidden Markov chain classification task

Here we give additional details on the hidden Markov chain (HMC) classification task.

### G.1  Task definition

Here we recall the task definition, providing a few additional details.

The $\mu$-th example in the dataset corresponds to an hidden Markov chain $q_1^\mu, \ldots, q_T^\mu$ of length $T = 30$, alternating between two hidden states, $q_t^\mu \in \{+, -\}$. The transition probability to the opposite state (" $\pm$ " $\rightarrow$ " $\mp$ ") is $p^\mu$. In other words, describing the "$\pm$" states as one hot vectors $+ : \begin{pmatrix} 1 \\ 0 \end{pmatrix}$ and $- : \begin{pmatrix} 0 \\ 1 \end{pmatrix}$, the transition probability matrix of the hidden Matkov chain is

$$\begin{pmatrix} 1 - p^\mu & p^\mu \\ p^\mu & 1 - p^\mu \end{pmatrix}.$$

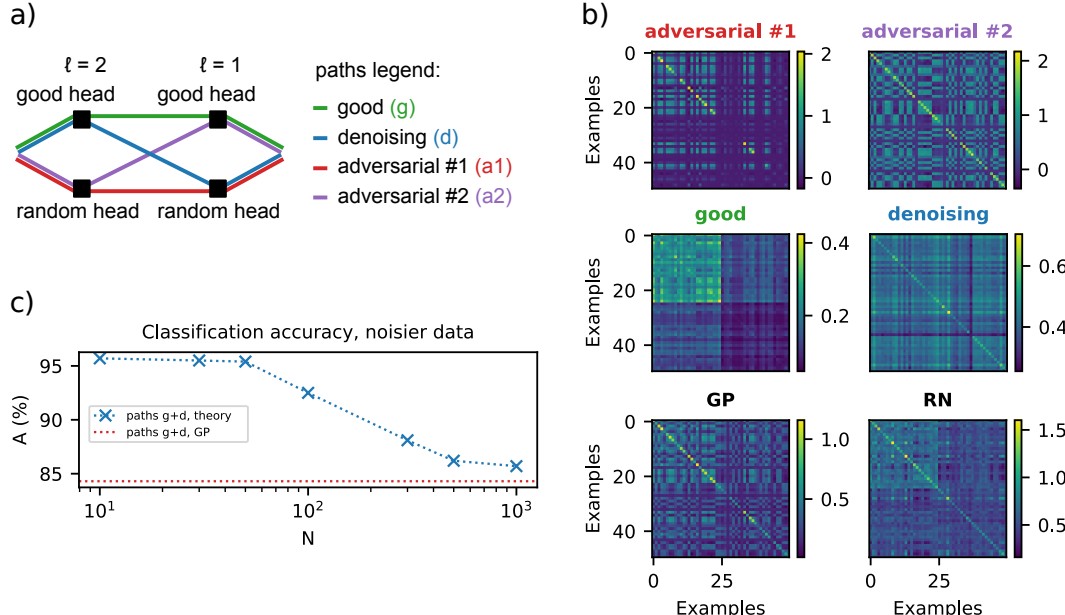

Figure 5: Hidden Markov chain task. **(a)** Schematics of the network and its attention paths. **(b)** Kernels. Same-path kernels associated with the 4 paths shown in (a), total kernel in the GP limit, and total kernel for $N = 10$ in the renormalized regime (RN). Examples on both the $x$ and $y$ axes are ordered by class (the first half correspond to the first class, the second half correspond to the second class). **(c)** Classification accuracy for a network consisting of only the good and denoising paths, $\sigma_\parallel = 1$ and $\sigma_\perp = 5$. The figure is analogous to Fig. 2(f), with the difference that we have replaced the random head involved in the denoising path with a uniform attention head.

The $\mu$-th chain can belong to one of two classes, labeled $y^\mu = \pm 1$, depending on whether $p^\mu = 0.3$ or $p^\mu = 0.7$ respectively.

The network is presented with visible states - the input tokens - which are a noisy, higher dimensional representation of the hidden states. These are given by

$$x_t^\mu = v_{q_t^\mu} + \eta_t^\mu$$

Here $v_\pm \in \mathbb{R}^{N_0}$ are two orthogonal feature vectors corresponding to the states "$\pm$". We set $N_0 = 200$ and

$$v_{+,i} = \begin{cases} \sqrt{2} & \text{if } i \leq 100 \\ 0 & \text{otherwise} \end{cases} \qquad v_{+,i} = \begin{cases} 0 & \text{if } i \leq 100 \\ \sqrt{2} & \text{otherwise} \end{cases}$$

The term $\eta_t^\mu$ is a zero-mean Gaussian noise, with $\langle \eta_t^\mu \eta_{t'}^{\mu'\,T} \rangle = \delta_{\mu,\mu'} \delta_{t,t'} (\sigma_\parallel^2 P_\parallel^\top \cdot P_\parallel + \sigma_\perp^2 P_\perp^\top \cdot P_\perp)$, where $P_\parallel$ and $P_\perp$ are the projectors along the subspace parallel or perpendicular to the plane spanned by $v_+$ and $v_-$.

We also preprocess the dataset in two ways: adding to each chain a beginning-of-sentence (bos) token of zeros at position $t = 0$, and concatenating to each token a one-hot positional encoding vector of size $T + 1$ (i.e. the number of tokens including the bos token).

We use $P = 100$ examples for training and $P^* = 1000$ examples for testing. For this task, we find the temperature $\mathcal{T}$ to not be relevant for improving the network performance. We therefore set it to a small, but finite value of $\mathcal{T} = 0.01$. A finite value of $\mathcal{T}$ is required to sample the predictor statistics, hence comparing the theoretical results with samples.

### G.2 Query and Key weights initialization

Here we give the details of the initialization of the fixed query and key weights.

We recall that we consider a network of $L = 2$ layers and $H = 2$ heads per layer, with readout from the first token (i.e., $t^* = 1$). The network has a total of 4 attention paths (Fig. 5(a)). For the first head

of each layer, we make a good choice of the fixed query and key weights, which defines a "good" attention path, achieving a good classification accuracy (cf. Sec. 4.1 in the main text). The remaining heads are initialized at random. Below we give the initialization details of these good and random heads.

Let us recall here the definition of the attention matrix $\Omega^{(\ell)h} \in \mathbb{R}^{T \times T}$ for head $h$ at layer $\ell$. Its matrix elements are defined as

$$\Omega_{st}^{(\ell)h} = \zeta \left( \frac{1}{N_0 \sqrt{G}} x_s^\top \cdot W_K^{(\ell)h\top} \cdot W_Q^{(\ell)h} \cdot x_t \right), \qquad W_Q^{(\ell)h}, W_K^{(\ell)h} \in \mathbb{R}^{G \times (N_0 + T + 1)}$$

where $s, t = 0, \ldots, T$, while $\zeta$ is the softmax function, applied along the direction of the token index $s$, and $G$ is the dimension of the query-key feature space. We directly initialize the query-key matrix product $W^{(\ell)h} := W_K^{(\ell)h\top} \cdot W_Q^{(\ell)h}$. Note that $W^{(\ell)h}$ is an $[N_0 + (T + 1)] \times [N_0 + (T + 1)]$ matrix, because we have appended a one-hot positional encoding vector to the $N_0$-dimensional input tokens (see Sec. G.1). In order to define the heads, it is convenient to decompose $W^{(\ell)h}$ into the block structure

$$W^{(\ell)h} = \beta \begin{pmatrix} W_{\text{ff}}^{(\ell)h} & W_{\text{fp}}^{(\ell)h} \\ W_{\text{pf}}^{(\ell)h} & W_{\text{pp}}^{(\ell)h} \end{pmatrix}$$

where $W_{\text{pp}}^{(\ell)h} \in \mathbb{R}^{(T+1) \times (T+1)}$ acts only on the one-hot positional encoding subspace, $W_{\text{ff}}^{(\ell)h} \in \mathbb{R}^{N_0 \times N_0}$ acts only on the subspace of the tokens' "features", and $W_{\text{fp}}^{(\ell)h}, W_{\text{pf}}^{(\ell)h\top} \in \mathbb{R}^{N_0 \times (T+1)}$ mix these two subspaces. The scalar $0 < \beta < \infty$ is a parameter controlling the "hardness" of the softmax function (for $\beta \to \infty$, the softmax becomes a hardmax). We set it to $\beta = 10$ for all heads.

### G.2.1 Good heads

Let us define the good heads, i.e. $W^{(\ell)h}$ for $h = 1$ and $\ell = 1, 2$. Note that the goal here is not to define heads that are provably good at solving the task, but rather to make a good guess for their initialization, based on our knowledge of the nature nature of the task.

**First layer.** For the head $h = 1$, $\ell = 1$ we define

$$W_{\text{ff}}^{(1)1} = \frac{1}{N_0} \left( v^+ - v^- \right) \left( v^+ - v^- \right)^\top, \qquad W_{\text{fp}}^{(1)1} = W_{\text{pf}}^{(1)1} = \mathbf{0}$$

and

$$\left[ W_{\text{pp}}^{(1)1} \right]_{tt'} = \frac{3}{2} \delta_{0,t} + 1 \delta_{t,t'+1} \qquad t, t' = 1, \ldots, T$$

where $\left[ W_{pp}^{(1)1} \right]_{tt'}$ is the component of $W_{pp}^{(1)1}$ at indices $t, t'$.

**Second layer.** The head $h = 1$, $\ell = 2$ implements uniform attention $\Omega_{st}^{(2)1} = \frac{1}{T+1}$, $\forall s, t = 0, \ldots, T$. It is defined by $W_{\text{ff}}^{(1)1} = W_{\text{fp}}^{(1)1} = W_{\text{pf}}^{(1)1} = \mathbf{0}$, and $W_{\text{pp}}^{(1)1} = \mathbf{1}$.

As discussed in the main text, the attention path defined by the good heads achieves a good classification accuracy. We can give an intuition as to why this is the case. Intuitively speaking, in the limit of a hardmax attention $(\beta \to \infty)$ and no noise $(\sigma_\parallel, \sigma_\perp \to 0)$, the attention path is "counting" the number of times a token has remained in the same state after a new step in the Markov chain. Indeed, the first head "detects" when there has not been a change of state between adjacent tokens. It does so by either attending nearby tokens if and only if they are in the same state, or attending "nothing" (in the sense of the zero beginning-of-sentence token). Then, the second head sums over the tokens attended by the first head, thereby "counting" the number of times a token has not changed state. More generally, outside the above mentioned limit, we can say that the good attention path focuses on the two most relevant pieces of information needed to solve the task: First, it preferentially pays attention to nearby tokens, which is important because of the memoryless nature of the Markov process; Second, it is able to detect the type of transition occurring between nearby tokens (i.e. remaining in the same state, or changing state), which is important to distinguish between the two classes, since they differ by their transition probability.

### G.2.2 Random heads

Let us define the random heads, i.e. $W^{(\ell)h}$ for $h = 2$ and $\ell = 1, 2$. These are initialized with Gaussian identically and independently distributed entries

$$\left[ W_{\mathrm{pp}}^{(\ell)h} \right]_{i,j} \sim \frac{1}{N_0} \mathcal{N}(0,1) \qquad \left[ W_{\mathrm{pp}}^{(\ell)h} \right]_{t,t'} \sim \mathcal{N}(0,1)$$

$$\left[ W_{\mathrm{pf}}^{(\ell)h} \right]_{t,j} \sim \frac{1}{\sqrt{N_0}} \mathcal{N}(0,1) \qquad \left[ W_{\mathrm{fp}}^{(\ell)h} \right]_{i,t'} \sim \frac{1}{\sqrt{N_0}} \mathcal{N}(0,1)$$

$\forall i, j = 1, \ldots N_0$ and $\forall t, t' = 0, \ldots, T$. Note that we take care of proper normalization of the above matrices, depending on which subspaces they act upon (i.e. the "features" or the "one-hot positions" subspaces).

As mentioned in the main text (Sec. 4.1), the random heads introduce three additional paths: two adversarial paths, deteriorating the performance of the good path, and one "denoising" path, improving the good path performance. We can get an intuition of why this is so by looking at their associated same-path kernels, Fig. 5(b). We can see that both the good-path and the two adversarial-path kernels appear very structured, with sharp excursions in their values for different pairs of examples. However, while the good-path kernel structure appears to be aligned with the task, well distinguishing the two classes, the adversarial-path kernels structure appears random w.r.t. to the task. We can expect that adding these adversarial kernels to the good one would destroy it's task-relevant structure, as can be visually understood from the total GP kernel. In contrast, the total renormalized kernel, in which the adversarial-path kernels do not contribute, preserves the task-relevant structure. Differently, the denoising-path kernel appears less structured and more uniform, with weaker noisy excursions. In fact, what we suspect is that there is nothing special about the specific realization of the random head involved in the denoising path, which is just implementing a noisy version of uniform attention. We verify this by substituting the random head with one implementing uniform attention and repeating the same experiment shown in Fig. 2(f) in the main text. This is shown in Fig. 5(c), were we plot the classification accuracy of the network consisting of the good and denoising paths alone, for the case of $\sigma_{\parallel} = 1$ and $\sigma_{\perp} = 5$. We can see that the results are completely analogous to those shown in Fig. 2(f) in the main text.

## H One-shot image classification task

### H.1 Training with gradient descent

Here we provide details of gradient descent training of our transformer-like model (Sec. 2) for the one-shot image classification task. We use the Omniglot dataset with the standard 1028/172/432-splits for the train/validation/test class splits [67] as implemented in `torchmeta` [68]. We use the Adam optimizer [63] using an initial learning rate of $3e^{-4}$ and a batch size of 128 for 10 epochs. We check the validation accuracy every 1000 steps and select the final model as the one that achieves the best validation accuracy. We use the binary regression loss as in the theory. Unlike in the theory, here we train all the model parameters including the key and query projection weight matrices. We set $N = 512$. All the models considered in this work can be trained on a single A100-40GB GPU within less than 2 hours.

### H.2 Additional results

Here we report further results on the one-shot image classification task.

### H.2.1 Optimal temperature

For the Bayesian model, we find tuning the Gibbs temperature $\mathcal{T}$ to be particularly important to optimally perform the task. All results for the network's classification accuracy presented in Sec. 4.2 and below are therefore shown at the optimal temperature for the given $N$, obtained by scanning the set of temperatures $\mathcal{T} \in \{a\, 10^{-b}, 1.0, 1.5\}$, where $a \in \{1, 2.5, 5, 7.5\}$ and $b \in \{1, 2\}$.

Note that temperature is optimized only for the in-distribution classification accuracy. In particular, we do not optimize for temperature when testing out-of-distribution, but rather keep the optimal temperature determined by evaluating the network in-distribution.

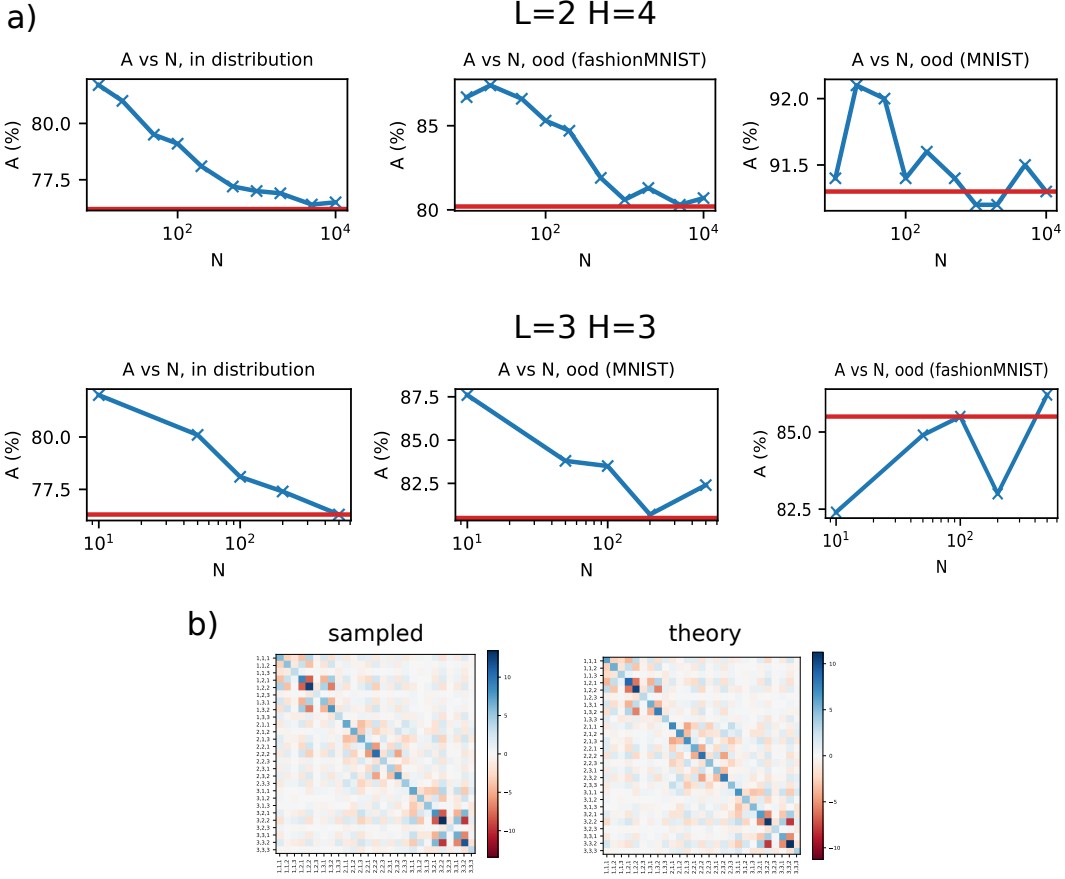

Figure 6: **(a)** Classification accuracy for varying $N$, tested both in-distribution (Omniglot) and out-of-distribution (MNIST, fashionMNIST). We report the results for the network architecture discussed in the main text ($L = 2$, $H = 4$) and a deeper one with one more layer ($L = 3$, $H = 3$). Theory: blue crosses, joined by blue curve. GP limit: red line. **(b)** Order parameter for the $L = 3$, $H = 3$ network, for $N = 10$. Similarly to the $L = 2$, $H = 4$ network shown in the main text, the order parameter showcases attention paths interplay, presenting strong off-diagonal elements that deviate from the GP limit.

For the network considered in the main text, we find the following optimal temperatures: $\mathcal{T} = 0.5$ for $N = 10, 20, 50, 100$; $\mathcal{T} = 0.25$ for $N = 200, 500, 1000$; $\mathcal{T} = 0.1$ for $N = 2000, 5000$; $\mathcal{T} = 0.075$ for $N = 10000$. Note that the optimal temperature grows consistently as $N$ becomes smaller. This can be readily understood by inspecting the equation for the mean predictor (Eq. 29), which we report here for convenience

$$\langle f^* \rangle = k^\top \cdot (K + \mathcal{T}\mathbb{I})^{-1} \cdot Y$$

where we recall that

$$K^{\mu\nu} = \frac{1}{H^L} \sum_{\pi,\pi' \in \Pi} U^{\pi\pi'} C^{\mu\nu}_{\pi\pi'} \qquad \mu, \nu = 1, \ldots, P$$

where $C_{\pi\pi'}$ is a path-path kernel. For decreasing $N$, we typically observe the order parameter growing in overall magnitude, which in turn affects the magnitude of the kernel $K$. As a consequence, also the optimal temperature needs to be rescaled. The fact that $U$ grows in magnitude for smaller $N$ can be understood from the energy term (Eq. 34) in the action (Eq. 32). We discussed in Sec. 3 that this can be seen as the negative log-likelihood of the labels vector $Y$ under a centered Gaussian distribution, whose covariance matrix is the kernel $K$. While the most effective way to minimize the energy term is that described in the main text (Sec. 3), i.e. aligning the kernel with the task, one more trivial way is to increase the log-likelihood variance in all directions (i.e. increasing the kernel's overall magnitude). In all of our experiments, we always observe this phenomenon of growing magnitude to a certain extent.

a)

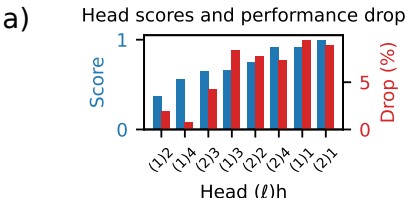

Head scores and performance drop

b)

accuracy after pruning

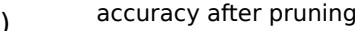

| # heads pruned | omniglot | fashion MNIST | MNIST |
|---|---|---|---|
| 0 | 83.8 | 86.6 | 86.6 |
| 1 | 81.8 | 88.2 | 88.2 |
| 2 | 80.2 | 88.9 | 88.9 |
| 3 | 71.6 | 82.3 | 58.8 |

c)

Head scores and performance drop

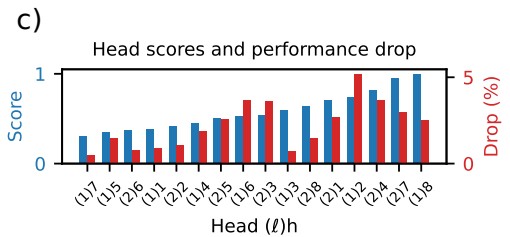

d)

accuracy after pruning

| # heads pruned | omniglot | fashion MNIST | MNIST |
|---|---|---|---|
| 0 | 86.0 | 79.8 | 91.9 |
| 1 | 85.5 | 81.0 | 91.1 |
| 2 | 84.6 | 79.0 | 90.8 |
| 3 | 83.7 | 83.1 | 90.3 |
| 4 | 82.4 | 77.9 | 90.9 |
| 5 | 80.4 | 80.6 | 87.6 |

Figure 7: Heads pruning experiment. **(a,c)** Head score (blue) and performance drop (red) after pruning the head, for the network trained with gradient descent. (a) smaller network considered in the main text ($L = 2$, $H = 4$); (c) larger network ($L = 2$, $H = 8$) **(b,d)** Classification accuracy of the model trained with gradient descent, after pruning a growing number of heads, in order of their head score. (b) smaller network considered in the main text ($L = 2$, $H = 4$); (d) larger network ($L = 2$, $H = 8$)

### H.2.2 Classification accuracy

We perform the same experiments on the classification accuracy for a deeper network ($L = 3$, $H = 3$). The results are shown in Fig. 6. Note that here we also report the test accuracy on MNIST, which was not shown in the main text. We can see that the results discussed in the main text are confirmed also for the deeper network. In particular, for the in-distribution classification accuracy, we consistently observe a performance improvement in the renormalized regime, with respect to the GP limit. When testing out of distribution, we can see that in the best cases (fashionMNIST for ($L = 2$, $H = 4$); MNIST for ($L = 3$, $H = 3$)), the performance improvement is preserved, or at the very worst (MNIST for ($L = 2$, $H = 4$); fashionMNIST for ($L = 3$, $H = 3$)) the GP and renormalized regime show comparable performance.

### H.2.3 Heads pruning

We repeat the head pruning experiment for a network with more heads per layer ($L = 2$, $H = 8$). We find that we can prune with marginal performance loss a similar percentage of heads as in the smaller network. The results are shown in Fig. 7. Again, we see that the head scores determined by our theory are qualitatively in line with the performance loss caused by pruning the corresponding head (Fig. 7(c)). Note that we do not expect a perfect alignment between the two quantities. The most important fact to verify is that the low scoring heads correspond to a small drop in performance. In Fig. 7(d) we show the classification accuracy after pruning an increasing number of heads, in order of their score. Up to 4 heads ($25\%$ of the network size) the in-distribution performance has only a marginal drop, identical to that obtained after pruning 2 heads in the smaller network considered in the main text (also accounting to $25\%$ of its size, Fig. 7(b)). We can also see that pruning up to three heads improves the out-of-ditribution performance on fashionMNIST, as we also observed for the smaller network.

