# OpenReview forum: "Dissecting the Interplay of Attention Paths in a Statistical Mechanics Theory of Transformers"
_NeurIPS.cc/2024/Conference — NeurIPS 2024 poster_

### Official Review · Reviewer_15FG · 2024-06-15

**Soundness:** 3
**Presentation:** 1
**Contribution:** 2
**Rating:** 5
**Confidence:** 3

**Summary:**

This paper proposes a way to analyze the interplay of attention paths. Statistical analysis is provided, and experiments are performed to verify the theory.

**Strengths:**

The idea considered in this paper is interesting, and the experiments seem to align with the theoretical justifications.

**Weaknesses:**

[1] This paper modifies the transformer model compared to the standard transformer. While changing the activation to linear seems to be acceptable for the ease of the derivation, the critical change in what is fed in the later layers seems strong: It significantly simplifies the format of equation (7) and the behavior of the considered transformer can be quite different from the standard transformer. The experiments are using two attention layers, which is significantly different from the real practice. It remains doubtful whether the analysis is still correct when using a deeper standard transformer.

[2] The writing of this paper can be improved in its theory parts: There is no formal theory statement, and readers have to trust the authors that we use equation (9) - (13) in the analysis. Without the formal theory statement, it is even more difficulty to track the derivations in the appendix: Readers have to go through every detail to understand what the authors want to do in (9) - (13).

[3] The writing of the paper can be improved in the experiment part: For example, it is not clear whether the authors implement a transformer using the considered architecture, or just a standard transformer. For image data, it is not clear whether additional embedding is used or not. In addition, the training details of the transformer is missing.

**Questions:**

Please use formal theoretical claim to express the key quantify of interest.

---

> ### Author Rebuttal · Authors · 2024-08-06
>
> We thank the reviewer for the precious feedback, which helped us to improve the communication of our findings.
>
> At the moment, however, we believe that our contributions have not been evaluated in the proper context.
>
> In particular, the reviewer seems to interpret our work as providing a method of analysis and its application. Instead, our work belongs to a more theory-focused line of research aimed at providing an analytical understanding of deep learning in the form of *exact analytical results*—in our case, an exact expression for the network's predictor on test examples.
>
> Before reading this response, we kindly ask the reviewer to read our global response, which should help in better contextualizing our work.
>
> 2. *[...] the critical change in what is fed in the later layers seems strong: It significantly simplifies the format of equation (7) and the behavior of the considered transformer can be quite different from the standard transformer. The experiments are using two attention layers, which is significantly different from the real practice. It remains doubtful whether the analysis is still correct when using a deeper standard transformer.*
>
> The reviewer is correct in identifying the simplifications present in our model with respect to the standard transformer, which we openly acknowledge in our manuscript. However, we believe these simplifications should be considered in the context of our theory-focused line of research. At the current state of the art of deep learning theory, simplifications need to be made, if exact analytical results are to be derived. In this context, our work actually takes important steps forward with respect to the simplified models considered in other comparable works (see global response for details). In particular, although clearly a simplification, our assumption on what is fed to the attention heads allows us to analytically tackle a multi-head multi-layer architecture, which remained inaccessible to previous works. This allows us to characterize a previously uncovered learning mechanism: attention paths interplay.
>
> A very relevant question is whether this mechanism would still be present, when the attention is fed with the layer’s preactivation, rather than the bare input. It is reasonable to expect that the mechanism would persist, but that it would be harder to disentangle from the learning of the attention itself, which would now also depend on the value weights. Future work could build upon our results to tackle this exciting mathematical challenge. This consideration will be included in the revised version of the manuscript, as per request of reviewer Y9xT.
>
> Regarding the experiments, again the rigorous theoretical framework limits the scale of testable networks. In the Bayesian framework, one needs to sample from the high dimensional posterior distribution of network weights, which is much more computationally costly than gradient descent, and technically challenging for a larger number of layers/heads. Large scale experiments are anyways out of scope for this context. Since, under the specified assumptions, our analytical results are exact for any depth, experiments are typically of small scale and serve two purposes: 1) To convince the reader, who may not want to check the rigorous derivations, that the analytical predictions are exact 2) To provide minimal, easily interpretable examples illustrating the theoretical insights. We also invite the reviewer to check the scale of experiments considered in other comparable works, which we cite throughout the introduction.
>
> 3. *There is no formal theory statement [...].*
>
> We thank the reviewer for their suggestion to improve the readability of our results. The revised manuscript will provide a formal theory statement. Currently, our theoretical results and related definitions are presented alongside their interpretation and implications for generalization. The revised manuscript will first present the results (eqs. 9-12) and assumptions in a compact and more formal statement, and only later discuss their interpretation.
>
> 4. *[...] it is not clear whether the authors implement a transformer using the considered architecture, or just a standard transformer. For image data, it is not clear whether additional embedding is used or not. In addition, the training details of the transformer is missing.*
>
> We thank the reviewer for their important suggestions to improve the presentation of our experiments.
> We always consider our transformer-like architecture, and never the standard transformer. This is because our work focuses on providing exact analytical results, rather than a method of analysis, so our experiments are focused on validating the theory and illustrating the predicted mechanisms on the considered architecture. The revised manuscript will make this clearer by explicitly referencing equations (1-4) defining the network, whenever it is mentioned in the experimental section.
>
> When nothing is said explicitly, the network is the one considered by the theory, i.e. “trained” by sampling its weights from the posterior distribution Eq. 8. The revised manuscript will state this explicitly. Details on the sampling method are given in “Experiments, Section F” in the Appendix. When the network is trained with gradient descent, we already state this explicitly and refer to  “Experiments, Section H.1” for training details.
>
> For images, the input tokens consist of just the normalized pixel values. No additional embedding is used. This will be stated explicitly in the revised text.
>
> We hope our response clarifies the scope of this work, and highlights our true contributions on both advancing the frontiers on the theoretical research on transformer-like models, and deep learning theory in general.  In our view, the current rating is unfair in light of these contributions.  If you find our response convincing/useful, please consider amending the score. Thank you very much.

---

> > ### Comment · Reviewer_15FG · 2024-08-07
> >
> > I appreciate the authors in preparing the rebuttal. Could you help provide a more detailed description for **"The revised manuscript will provide a formal theory statement."** It would be great to have the statement here.
> >
> > Thanks.

---

> ### Author Response · Authors · 2024-08-09
> **Introductory comment to the theory statement**
>
> We thank the reviewer for giving us the opportunity to provide the
> formal theory statement, which we could not include in our original
> response due to length constraints. The theory statement will appear in the next comment. As we stated in the rebuttal, all parts of the manuscript about the
> interpretation and implications of our results will be put after this
> theory statement. Please refer to the submitted manuscript
> for any referenced equations which do not appear in the statement itself (i.e. Eqs 1-8, which will appear in the manuscript before the theory statement).
> Also note that we made the following changes in notation:
> - 1. We renamed
> the query and key weights from $\textbraceleft Q ^{\left(\ell\right)h},K{} ^{\left(\ell\right)h}\textbraceright  _{\ell,h=1} ^{L,H}$
> to $\textbraceleft W _{Q} ^{\left(\ell\right)h},W _{K} ^{\left(\ell\right)h}\textbraceright  _{\ell,h=1} ^{L,H}$,
> in order to avoid confusion with the kernel $K$.
> - 2. We renamed the
> network predictor on a test example from $f ^{P+1}$ to $f ^{*}$.
> - 3. We renamed the network inputs from $x ^{\left(0\right)}$ to $x$.
>
> This new naming scheme will be consistently implemented throughout
> the revised manuscript.

---

> ### Author Response · Authors · 2024-08-09
> **Theory statement**
>
> ### Theory statement ###
>
> **Definitions.** Consider a training dataset consisting of $P$
> inputs $x ^{\mu}\in\mathbb{R} ^{N _0\times T}$ and associated labels
> $y ^{\mu}\in\mathbb{R}$, where $\mu=1,\ldots P$. Call $X\coloneqq \textbraceleft x ^{\mu} \textbraceright _{\mu=1} ^{P}$
> the set of training inputs and $Y\in\mathbb{R} ^{P}$ the vector of
> training labels with $\mu$-th component $y ^{\mu}$. Consider a network
> defined by Eqs. (1-4) and in particular call $f ^*$ the network
> output (Eq. 4) corresponding to a test input $x ^* \in\mathbb{R} ^{N _0\times T}$.
> We remind the reader of the following network hyperparameters: the
> input's embedding dimension $N _0$, the hidden layer's width $N$,
> the number of tokens $T$, the number of attention heads per layer
> $H$, and the number of attention layers $L$.
>
> **Assumptions.** Assume the network weights $\Theta\coloneqq\left(V ^{\left(0\right)},\textbraceleft V ^{\left(\ell\right)h}\textbraceright  _{\ell,h=1} ^{L,H},a\right)$
> are distributed according to the Bayesian posterior distribution defined
> in Eq. 8, with temperature $\mathcal{T}>0$, while the query and key
> weights $\textbraceleft W _{Q} ^{\left(\ell\right)h},W _{K} ^{\left(\ell\right)h}\textbraceright  _{\ell,h=1} ^{L,H}$
> are fixed.
>
> Assume $N,N _0,P\to\infty$, with $P/N\coloneqq\alpha\in\mathbb{R} ^{+}$
> and $P/(N _0H ^{L})\coloneqq\alpha _0\in\mathbb{R} ^{+}$, where $\alpha$,
> $\alpha _0$ as well as other size parameters $T,H,L\in\mathbb{\mathbb{N}}$
> are finite.
>
> **Claim.** Under the above assumptions,
>
> (1) the mean predictor under the posterior distribution (Eq. 8) is
> given by
> $$\mathbb{E}\left[f ^{*}\right]=k ^{\top}\cdot\left(K+\mathcal{T}\mathbb{I}\right) ^{-1}Y, \qquad (9) $$
> where the average is w.r.t. to the posterior distribution (Eq. 8).
>
> The vector $k\in\mathbb{R} ^{P\times1}$ and the matrix $K\in\mathbb{R} ^{P\times P}$
> are defined in terms of a kernel function $\mathcal{K}:\mathbb{R} ^{N _0\times T}\times\mathbb{R} ^{N _0\times T}\to\mathbb{R}$
> as $k ^{\mu}\coloneqq\mathcal{K}\left(x ^{*},x ^{\mu}|U\right)$ and
> $K ^{\mu\nu}\coloneqq\mathcal{K}\left(x ^{\mu},x ^{\nu}|U\right)$, for
> $\mu,\nu=1,\dots,P$. The kernel function is given by
> \begin{equation}
> \mathcal{K}\left(x,x'|U\right)=\frac{1}{H ^{L}}\sum _{\pi,\pi'\in\Pi}U ^{\pi\pi'}C _{\pi\pi'}\qquad\mathrm{with}\qquad C _{\pi\pi'}\coloneqq\frac{1}{N _0}\xi ^{\pi}\left(x\right) ^{\top}\cdot\xi ^{\pi'}\left(x'\right)\, \qquad (10)
> \end{equation}
> where $\xi ^{\pi}\left(x\right)$ is the ``attentioned input'' corresponding
> to an input $x\in\mathbb{R} ^{N _0\times T}$, along path $\pi\in\Pi$
> (Eq. 7) and $\Pi$ is the set of all attention paths for a given architecture
> with $|\Pi|=H ^{L}$.
>
> The matrix $U\in\mathbb{R} ^{H ^{L}\times H ^{L}}$, called *order
> parameter*, is a positive semi-definite matrix given by
> \begin{equation}
> U=\underset{\tilde{U}}{\mathrm{argmin}} \ S(\tilde{U};X,Y)\, \qquad (11)
> \end{equation}
> where the scalar function $S$ called the *action* is defined
> as
> \begin{equation}
> S(U;X,Y)=\mathcal{L}(U)+\alpha\mathcal{E}(U;X,Y)\, \qquad (12)
> \end{equation}
> The scalar function $\mathcal{\mathcal{E}}$, which we call the *energy*,
> is given by
> \begin{equation}
> \mathcal{E}(U;X,Y)=\frac{1}{P}\ln\det\left(K(X,X|U)+\mathcal{T}\mathbb{I}\right)+\frac{1}{P}Y ^{\top}\cdot\left(K(X,X|U)+\mathcal{T}\mathbb{I}\right) ^{-1}\cdot Y, \qquad (13)
> \end{equation}
> where $K\coloneqq K(X,X|U)$ is the $P\times P$ training kernel matrix,
> defined according to Eq. (10). The expression for the scalar function
> $\mathcal{L}$, which we call *entropy*, is lengthy and is given
> in Appendix B.1.
>
> (2) In the particular case of a single head per layer $H=1$, $U$ is a scalar, and the entropy assumes the simple form $\mathcal{L}\left(U\right)=\sigma ^{-2\left(L+1\right)}U-\ln\left(U\right)$,
> where $\sigma ^{2}$ is the variance of the Gaussian prior on the network
> weights $\Theta$ (see Eq. 8).
>
> (3) For general $H$, $\mathcal{L}\left(U\right)$ is minimized by
> $U ^{\pi\pi'}=\sigma ^{2\left(L+1\right)}\delta _{\pi,\pi'}$,
> which therefore is always the solution of Eq. (11) in
> the GP limit defined by $\alpha\to0 ^{+}$.
>
> (4) The matrix $U$ obeys the following relation
> \begin{equation}
> U ^{\pi\pi'}=\frac{1}{N}\mathbb{E}[V _{\text{eff}} ^{\pi}\cdot V _{\text{eff}} ^{\pi'\top}], \qquad (14)
> \end{equation}
> where $V _{\text{eff}} ^{\pi}\in\mathbb{R} ^{1\times N}$ are the effective weights along path $\pi$ (Eq. 6).
>
> **Derivation:** See Appendix C.

---

> > ### Comment · Reviewer_15FG · 2024-08-09
> >
> > I appreciate the authors in providing the detailed revised material for the theory. I still feel that it is not sth as we usually refer to as a "theoretical statement". However, if treating the results as a new analysis tool, it is an interesting contribution. I have raise my score to 5.

---

> ### Author Response · Authors · 2024-08-10
>
> Thank you very much for taking the time to reassess the value of our contribution and for raising the score. While we hoped for a higher score, we respect your perspective and are willing to further improve our theoretical statement based on your feedback.
>
> We understand that the reviewer may be referring to theoretical statements in the field of mathematics. In this case, we would like to clarify that our work adopts the methods and presentation style of theoretical physics, according to which our results have the validity of a predictive theory, rather than a mere analysis tool.
>
> Theoretical physics has a long history in developing theories of artificial neural networks [1-5], whose contribution has always been of high relevance to conferences like NeurIPS (see, e.g., [6-11]). The field of theoretical machine learning is growing quickly, with different methods from mathematics, physics, and computer science, and each discipline has its own distinct goals and ways of communicating results.
>
> We recognize the importance of making our work accessible to researchers across these disciplines and are committed to refining our presentation in this direction. We welcome any further feedback to help us improve our work.
>
> ### References ###
> [1] Hopfield, J. J. Proc. Natl Acad. Sci. USA 79, 2554–2558 (1982)
>
> [2] Amit, D. J., Gutfreund, H. & Sompolinsky, H. Phys. Rev. Lett. 55, 1530–1533 (1985)
>
> [3] Gardner, E. J. Phys. A 21, 257–270 (1988)
>
> [4] Gardner, E. & Derrida, B. J. Phys. A 21, 271–284 (1988).
>
> [5] For a review on recent works, see, e.g.: Bahri, Y. et al. Ann. Rev. Cond. Matt. Phys. 11, 501–528 (2019)
>
> [6] Krogh, Anders, and John Hertz. Advances in neural information processing systems 4 (1991).
>
> [7] Cortes, Corinna, et al. Advances in neural information processing systems 6 (1993).
>
> [8] Saxe, A., McClelland, J., & Ganguli, S. (2014). Proceedings of the International Conference on Learning Represenatations 2014.
>
> [9] Bordelon, Blake, Abdulkadir Canatar, and Cengiz Pehlevan. International Conference on Machine Learning. PMLR, 2020.
>
> [10] Gerace, Federica, et al.  International Conference on Machine Learning. PMLR, 2020.
>
> [11] Lee, Jaehoon, et al. Advances in neural information processing systems 32 (2019).

---

### Official Review · Reviewer_Y9xT · 2024-07-12

**Soundness:** 3
**Presentation:** 2
**Contribution:** 3
**Rating:** 7
**Confidence:** 4

**Summary:**

The paper investigates Bayesian learning of the value weight matrices of a deep multi-head attention network without MLPs employing the back-propagating kernel renormalization (BPKR) [1] technique in the linear regime, where the training set size $P$ scales with the width $N$, i.e., $P/N = O(1)$. It finds that the network’s kernel is a sum of constituent kernels operating on different pairs of attention paths, rescaled based on their alignment with the target task. This renormalization enhances generalization compared to previously studied infinite-width (GP) limits. The paper validates its theoretical predictions through experiments on a synthetic task and in-context classification with simple image datasets, finding qualitative agreement with gradient-based training. Additionally, it shows that the theory’s predictions can be used to prune less relevant attention heads without significantly impacting performance.

[1] Li, Q. and Sompolinsky, H., 2021. Statistical mechanics of deep linear neural networks: The backpropagating kernel renormalization. Physical Review X, 11(3), p.031059.

**Strengths:**

- **Originality:** The application of BPKR to attention models and the characterization of the network’s kernel as a task-relevant weighted sum of path-path kernels is novel.
- **Quality:** The statistical mechanics analysis is technically sound and validated with comprehensive numerical experiments on both synthetic and real data, demonstrating the robustness of the findings. The paper provides the code to reproduce its empirical results.
- **Clarity:** The paper is generally clear, although there are areas for improvement (see Weaknesses section).
- **Significance:** The paper considers simplified transformer architectures, which remain challenging for theoretical approaches. The developed theory extends beyond the GP limit, predicting task-adaptivity properties that are necessary to explain the success of modern learning algorithms and are not captured by infinite-width limits. While the insights are intuitive and may not significantly enhance the understanding of transformers or their success, the theory is quantitative and offers non-trivial generalization predictions.

**Weaknesses:**

1. **Strong assumptions:** The analysis relies on several very strong assumptions: (i) linearity of the network output in the value weights, (ii) applying the attention at any depth on the network input, (iii) considering frozen (and already learned) query and key matrices, and (iv) a system at equilibrium (i.e., a Gibbs distribution over the parameters of the model). These assumptions clearly limit the relevance of the results.  Although some of the empirical results seem to suggest that the last two assumptions can be relaxed, the impact of (i) and (ii) on the conclusions remains unclear. It would be beneficial to discuss the implications of these assumptions further.
2. **Background:** The paper is dense and lacks sufficient background material to aid the reader in following it and understanding where the results come from. The BPKR technique and its assumptions and rationale are not adequately introduced. Additionally, the paper mentions the "network’s kernel" early on without really specifying which object it is considering (as there are various kernels in deep learning, e.g., NTK etc.). Providing a more detailed introduction and some intuition on BPKR, along with its assumptions and high-level steps in the main manuscript, would significantly improve clarity.
3. **Clarity and completeness:** Sec. 4.1.1 is not very clear and easy to follow. It would be helpful to explain the structure of the different heads within the main text, rather than repeatedly directing the reader to long appendices.
4. **Typos:** The manuscript contains several typos (“thermodinamic” in line 106, “generalizaton” in line 147, “task-specifc” in line 161, “cathegorized” in line 241, “taks” in line 262, “on this regard” in line 330).

**Questions:**

5. If I understand correctly, unlike the dense case, thanks to the transformer architecture, the kernel is not rescaled by a scalar renormalization variable, resulting in more interesting outcomes compared to [1]. Can you elaborate more on this?
6. In Fig. 3 (d), is the GD map obtained for the same network from which the trained query and key weights were obtained for the theory?
7. In general, in your experiments, do you always apply attention to the input at all depths of the network?
8. What do you mean by “efficiently” (line 57)?

[1] Li, Q. and Sompolinsky, H., 2021. Statistical mechanics of deep linear neural networks: The backpropagating kernel renormalization. Physical Review X, 11(3), p.031059.

**Limitations:**

The paper adequately lists its limitations in Section 5. I do not foresee any potential negative societal impacts arising from this study.

---

> ### Author Rebuttal · Authors · 2024-08-06
>
> We thank the reviewer for their valuable time reviewing our work and for many positive comments. We’d like to address the reviewer’s remaining concerns as follows.
>
> 1. *The analysis relies on several very strong assumptions: (i) linearity of the network output in the value weights, (ii) applying the attention at any depth on the network input, [...] the impact of (i) and (ii) on the conclusions remains unclear. It would be beneficial to discuss the implications of these assumptions further.*
>
> We agree with the reviewer that further discussion on assumptions (i) and (ii) would be beneficial.
>
> Regarding assumption (i), we discuss in the paper a potential way forward, following the heuristic arguments in [Li, Sompolinsky, PRX, 2021]. There, the analytical results for a deep linear network are heuristically extended to a network with ReLU activations, by replacing the linear GP kernels with the ReLU GP kernels. Despite having a more complex renormalization, also in our case the renormalized kernel can be seen as a linear combination, weighted by the order parameter, of many “GP” kernels, i.e., the path-path kernels. Therefore, we foresee how a similar argument could potentially be applied in our case. This suggestion, which was briefly mentioned in lines 330-332, will be discussed in more detail in the revised manuscript.
>
> Assumption (ii) is indeed a limitation of our model, when compared to the standard transformer. However, we would like to emphasize how this assumption allows us to consider an architecture featuring both multiple heads and multiple layers, enabling the characterization of attention paths interplay. The question is whether such a mechanism would still be present when relaxing assumption (ii), which seems reasonable to believe. The theoretical challenge, however, would be to disentangle the learning of attention paths interplay from the learning of the attention paths themselves, because now also the attention matrix would depend on the value weights. This discussion will be included in the revised manuscript. If the reviewer is interested, we provide further details in our response to reviewer 9bWs.
>
> 2. *If I understand correctly, unlike the dense case, thanks to the transformer architecture, the kernel is not rescaled by a scalar renormalization variable, resulting in more interesting outcomes compared to [1]. Can you elaborate more on this?*
>
> The reviewer is perfectly right: the multi-head transformer architecture induces a non-trivial renormalization encoded in the matrix order parameter. This results in a crucial improvement in performance. Indeed, for deep linear fully connected architectures, where the order parameter is purely scalar, the renormalization only affects the variance of the predictor, while the mean predictor remains the same as in the GP limit. Conversely, a matrix kernel renormalization produces a change in the mean predictor, thereby improving its generalization performance with respect to the GP limit. A matrix order parameter cleverly combines specific architectural components – the attention paths in our case  – offering deeper insight into the critical role of architecture in finite width networks.
>
> We kindly invite the reviewer to also evaluate the importance of the above contribution for the theory of deep learning in general, independently of the transformer’s context. We elaborate on this point in our global response.
>
> 3. *[...] The BPKR technique and its assumptions and rationale are not adequately introduced [...] the paper mentions the "network’s kernel" early on without really specifying which object it is considering [...].*
>
> We thank the reviewer for this precious feedback, which helps us to improve the communication of our findings. The new version of the manuscript will integrate the reviewer’s suggestions by:
> - a. Specifying to which kernel we refer more precisely.
> - b. Outlining the rationale behind the BPKR method: explain how the expectation under the Gibbs distribution of network weights can be reduced to an expectation over a lower-dimensional distribution of macroscopic order parameters, which is obtained by gradual integration of the weights; explain how the so-reduced expectation can then be solved via the saddle-point method in the thermodynamic limit.
> - c. Clarify the significance of the thermodynamic limit assumptions.
>
> 4. *It would be helpful to explain the structure of the different heads within the main text, rather than repeatedly directing the reader to long appendices.*
>
> We welcome the reviewer’s suggestion to improve the clarity of Section 4.1.1. The revised manuscript will contain an explanation of the key features of the different heads, while keeping the mathematical definitions in the appendix. In particular, we will explain how the first good head makes use of the Markov nature of the task by attending only to nearby tokens and checking whether they match, while the second good head performs uniform attention.
>
> 5. *In Fig. 3 (d), is the GD map obtained for the same network from which the trained query and key weights were obtained for the theory?*
>
> Yes. The revised text will explain this more clearly.
>
> 6. *In general, in your experiments, do you always apply attention to the input at all depths of the network?*
>
> The attention matrix $\Omega^{(\ell)}$ is always a function of the bare input $x^{(0)}$, irregardless of the layer $\ell$ (see Eq. 3). The attention matrix is however applied to the preactivation to that layer $x^{(\ell)}$ (see Eq. 2).
>
> 7. *What do you mean by “efficiently” (line 57)?*
>
> We mean that we prune the less relevant heads so that the loss in performance is minimal. We have rewritten the sentence as follows: “we show that a trained network can be reduced in size with minimal performance loss.”
>
> We hope our response provides clarifications to all the concerns the reviewer has raised. If you find our response convincing/useful, please consider increasing the score. Thank you very much.

---

> > ### Comment · Reviewer_Y9xT · 2024-08-13
> >
> > I thank the authors for their detailed answers to the reviews, which addressed my initial comments. I personally appreciate the theoretical physics approach of the work, particularly the formulation of the transformer kernel as a task-dependent combination of path-path kernels. Furthermore, I recognize that making (sometimes strong) assumptions is necessary to advance our theoretical understanding of deep learning systems. With the proposed changes, I recommend acceptance and have adjusted the score accordingly. I encourage the authors to include all the new discussions in the revised version.

---

> > > ### Author Response · Authors · 2024-08-13
> > >
> > > Thank you very much for your response and the updated score. We are glad to hear that the reviewer found our response useful. We will make sure to include these discussions in the final version.

---

### Official Review · Reviewer_9bWs · 2024-07-12

**Soundness:** 4
**Presentation:** 4
**Contribution:** 4
**Rating:** 8
**Confidence:** 4

**Summary:**

This paper provides an interpretation of wide (large embedding dimension, $N$) multi-head attention-only transformers solving in-context learning tasks as performing high-dimensional kernel combining. The paper derives the exact statistics of the predictor, when the number of training examples $P$, and $N \to \infty$, but $P/N \to \mathcal{O}(1) > 0$. The main result is the revelation that the predictor statistics can be expressed as the weighted sum of independent kernels, each pairing two attention paths through the multiple heads across the layers, and the weights depend on the relevance of the path to the task to be solved. Capitalising on this result, the paper proposes a network size-reduction method by efficient pruning of certain the attention paths with marginal performance loss.

**Strengths:**

(1) The paper introduces a novel way of looking at generalization capabilities of a transformer through the lens of the statistical mechanics theory in finite thermodynamic limit.
Specifically, the result shows that transformer-like architectures perform a task-relevant kernel combining, where each kernel is based on a path through the multi-head, multi-layer network. This is a crucial direction that offers valuable insights into the nature of the attention mechanisms.

(2) The proofs are clearly written with the necessary steps (barring some steps like saddle-point methods) to follow.

(3) The experimental results (Figure 3) greatly help in understanding the theoretical results (Section 3) better.

(4) The contents and organisation of the paper is easy to understand.

**Weaknesses:**

(1) One minor weakness is that the experiments are limited to binary classification.

**Questions:**

(1) I am curious about the authors' thoughts on what can happen when the attention is computed from the outputs of the previous layer and not always from the bare inputs?

(2) In order to prune the attention paths, the order parameters $U_{\pi, \pi'}$ (Eq. (13)) must be computed, which involves averaging over the Gibbs distribution (Eq. (8)). How can this be computed in real architectures?

**Limitations:**

(1) The limitations are already stated towards the end of the paper; (i) it does not consider the learning of query-key matrices, which is a crucial step in functioning of the attention mechanisms (ii) the attention is always computed from the bare inputs, which limits the applicability of the theory to practical transformers. However, these do not reduce the value of the contribution of the paper.

---

> ### Author Rebuttal · Authors · 2024-08-06
>
> We thank the reviewer for their valuable time reviewing our work, for the many positive comments, and for the intriguing questions.
>
> Below are our replies to the reviewer’s questions, as well as minor concerns.
>
> 1. *One minor weakness is that the experiments are limited to binary classification.*
>
> We agree that considering a network with multiple outputs would be interesting, and readily implementable within our theory. We would like to share with the reviewer why we actively decided to not include these kinds of experiments in this first work exploring our theory. The renormalization effect coming from having a number M of multiple outputs is the same for any architecture (e.g. the same as in deep linear networks [Li, Sompolinsky, PRX 2021]), and results in the order parameter gaining an additional pair of “output indices” ranging from 1 to M. In this work, we wanted to focus on illustrating exclusively the renormalization effects that are proper of the transformer architecture—-that of attention paths combination. Exploring the interplay between the two renormalization effects (multiple outputs and multiple paths) could be the topic of future work, with potentially exciting applications to language and next-token prediction tasks. One caveat in this context, is that one would need to be careful of a too large number of outputs M, which may spoil the thermodynamic limit which assumes M finite.
>
> 2. *I am curious about the authors' thoughts on what can happen when the attention is computed from the outputs of the previous layer and not always from the bare inputs?*
>
> It seems reasonable to believe that an attention paths combination mechanism would still persist in this case. The theoretical challenge, however, would be to disentangle the learning of attention paths interplay from the learning of the attention paths themselves. Indeed, when the attention is a function of the bare input, our results for the renormalized kernel
> $$K=\sum_{\pi,\pi'\in\mathrm{\Pi}}U^{\pi\pi'}C_{\pi\pi'}$$
> show that the value weights’ statistics enter only in determining the order parameter $U$, that is the interplay between attention paths. If the attention were to be a function of the previous layer’s output, instead, also the attention path kernels $C$ would depend on the value weights, which in this case affect the learning of the attention paths themselves. In other words, the value weights would influence performance in two key ways: (i) by optimizing the representation of inputs to the attention heads, and (ii) by improving the recombination of attention paths.
>
> Obtaining analytical progress in this scenario is challenging, primarily due to the introduction of non-linearities in the trained weights through the attention’s softmax. A first step to advance in this computation could be to consider linear attention heads, which may simplify the analysis.
>
> This discussion will be included in the revised manuscript, as also prompted by a question from reviewer Y9xT.
>
> 3. *In order to prune the attention paths, the order parameters 𝑈𝜋,𝜋′  (Eq. (13)) must be computed, which involves averaging over the Gibbs distribution (Eq. (8)). How can this be computed in real architectures?*
>
> We would like to be sure to consider the case meant by the reviewer with “real architectures”, so let us consider a few scenarios below:
>
> - a. For our model in the Bayesian framework, the order parameter can be equivalently computed by either solving the self-consistent equation (as in figure 3d “theory”), or sampling the posterior (as in figure 3d “sampled”).
> - b. For our model trained with gradient descent, one can just take the learned query and key weights, and plug them into procedure (a) to obtain the order parameter for the value weights. Another more empirical approach that we did not try, would be to get multiple “samples” of the value weights by retraining the network with gradient descent, but for fixed query and key weights coming from the first iteration of the training (this is somehow what we did to compute the order parameter in figure 3d “gradient descent”, only that there we used a single “sample”).
> - c. For the more standard transformer with attention as a function of the bare input and trained with gradient descent, one could still do something similar to (b). The difference would be that now also the attention matrix $\Omega$ would depend on the value weights. Instead of taking the learned query and key weights, and plugging them into procedure (a), one could just take the entire attention matrix $\Omega$ (as computed on each example from the network's activations) and plug it into procedure (a).
> - d. If we also add nonlinear MLP blocks in between each head, one could still apply the same procedure as in (c),  but, when arriving to step (a), he could only proceed by sampling the posterior, while the option of solving the self-consistent equation would not be available due to the self-consistent equation not being defined for the nonlinear case. A possible way to define the self-consistent equation could be to heuristically extend it to the nonlinear case, using the same heuristic argument as in [Li, Sompolinsky, PRX, 2021]. There, the analytical results for a deep linear network are heuristically extended to a network with ReLU activations, by replacing the linear GP kernels with the ReLU GP kernels. Despite having a more complex renormalization, also in our case the renormalized kernel can be seen as a linear combination, weighted by the order parameter, of many “GP” kernels, i.e., the path-path kernels. Therefore, we foresee how a similar argument could potentially be applied in our case.
>
> 4. *...barring some steps like saddle-point methods*
>
> The revised manuscript will have an improved explanation of the saddle point technique. We ask the reviewer, if possible, to help us by further elaborating on the unclear points.
>
> Once again, we thank the reviewer for this thorough review and the very positive rating.

---

> ### Author Response · Authors · 2024-08-12
>
> We noticed a typo in our reply, point 3.c :
>
> "For the more standard transformer with attention as a function of *the bare input*" -> "For the more standard transformer with attention as a function of *its layer's preactivations*".

---

### Official Review · Reviewer_5PZg · 2024-07-13

**Soundness:** 3
**Presentation:** 2
**Contribution:** 2
**Rating:** 3
**Confidence:** 3

**Summary:**

This work studies the mechanism of attention in deep Transformers. The theory shows that the prediction process can be expressed as a combination of kernels of different attention paths. Experiments are conducted to verify the theory, which also motivates a pruning method on different attention heads.

**Strengths:**

1. The mechanism of the attention path is interesting and reasonable.
2. The theory and experiments support each other and make sense.

**Weaknesses:**

1. The experiments are simple. It will be better to verify the mechanism on larger datasets, larger models, and more complicated tasks. Experiments use Transformers of 2 or 3 layers, which are not deep. However, the abstract claims a "deep multi-head self-attention network".

2. The presentation is somehow weird. The order of figures in Figures 2 and 3 is strange. Also, Figures 2 and 3 can be divided into multiple figures, since the experiments in these subfigures are about different things.

3. Some references on the theoretical mechanism of Transformers in learning are missing.

Olsson et al., 2022. In-context Learning and Induction Heads.

Li et al., ICML 2024. How Do Nonlinear Transformers Learn and Generalize in In-Context Learning? (This work also includes model pruning based on the mechanism of Transformers)

Nichani et al., ICML 2024. How Transformers Learn Causal Structure with Gradient Descent.

Reddy et al. ICLR 2024. The mechanistic basis of data dependence and abrupt learning in an in-context classification task.

**Questions:**

1. Can you have some discussion about the following related work about model pruning? This work also finds that some heads can be pruned during the inference. However, their experiments are implemented by real-world state-of-the-art LLMs. Their results are more complete in terms of performance, efficiency, and on more challenging tasks.

Liu et al., ICML 2023. Deja Vu: Contextual Sparsity for Efficient LLMs at Inference Time.

**Limitations:**

There is no potential negative societal impact of their work. The extension of the proposed method and analysis to larger models is the biggest concern.

---

> ### Author Rebuttal · Authors · 2024-08-06
>
> We thank the reviewer for the precious feedback, which helped us to improve the communication of our findings.
>
> At the moment, however, we believe that our contributions have not been evaluated in the proper context. Before reading this response, we kindly ask the reviewer to read our global response, which should help in better contextualizing our work.
>
> 1. *The experiments are simple. It will be better to verify the mechanism on larger datasets, larger models, and more complicated tasks. Experiments use Transformers of 2 or 3 layers, which are not deep. However, the abstract claims a "deep multi-head self-attention network".*
>
> As explained in the global response, our work is theory-focused and provides exact analytical expressions for the network predictor. In this context, experiments are typically designed to serve two purposes: 1) To convince the reader, who may not want to check the details of the theoretical derivations, that the analytical predictions are exact. 2) To provide minimal, easily interpretable examples illustrating the theoretical insights. Specifically, we focused on experiments with only 2 or 3 layers and 2 to 4 heads because the resulting order parameter remained visually interpretable in that case.
>
> In the Bayesian framework adopted here, numerically validating our derivations on larger models and datasets would require an exponentially larger computational cost—much larger than with gradient descent—other than being technically challenging, because it requires sampling the network weights from an high dimensional posterior distribution (Eq. 8). We believe this to be out of the scope of our work: our theoretical results are rigorous and apply for any depth, under the thermodynamic limit specified by the theory.
>
> We acknowledge that the term “deep” is used in the literature with varying meanings. In the context of theory, where analytical results have predominantly been obtained for single-layer attention, we followed the convention of using “deep” to emphasize that our derivations apply to models with an arbitrary number of layers. However, if the reviewer believes it is more appropriate, we are willing to change the term “deep” in the abstract to “multi-layer.”
>
> 2. *The presentation is somehow weird. The order of figures in Figures 2 and 3 is strange. Also, Figures 2 and 3 can be divided into multiple figures, since the experiments in these subfigures are about different things.*
>
> We thank the reviewer for spotting this problem. We have changed the order in both figures and made them more readable. The updated figures are available in the joint PDF. We have decided not to split the figures, because all the subplots are closely related and together show how the path combination mechanism affects the performance. For instance, in order to understand both the accuracy trend and the pruning mechanism in Fig. 3, it is useful to look at the structure of the order parameter.
>
> 3. *Some references on the theoretical mechanism of Transformers in learning are missing.*
>
> We thank the reviewer for pointing us towards these missing references. These will appear in the revised version as follows. Refs. [Li et al., ICML 2024; Nichani et al., ICML 2024] will be included in the Introduction, in the discussion of other theory-focused, closely related works. Refs. [Olsson et al., 2022; Reddy et al. ICLR 2024] are also highly relevant, but more suitable for the Discussion section, since they focus on empirical methods [Olsson et al., 2022] and phenomenological models [Reddy et al. ICLR 2024] that are more distant from the focus of our work on exact derivations of the network’s predictor.
>
> 4. *Can you have some discussion about the following related work about model pruning? This work also finds that some heads can be pruned during the inference. However, their experiments are implemented by real-world state-of-the-art LLMs. Their results are more complete in terms of performance, efficiency, and on more challenging tasks.*
>
> We thank the reviewer for pointing us towards this interesting work [1], which will be discussed in the Discussion section of the revised manuscript.
>
> However, we would like to stress that paper [1] is very different in scope and methodology to our work. Our work is theory-focused, and aims at providing exact analytical expressions that describe the network’s learning mechanisms. The cited work [1] is application-focused, and aims at providing a competitive pruning algorithm. In our work, the application to head pruning is neither the main result nor the focus, and we are not making any claim about it being competitive: we only use it to elucidate the kernel combination mechanism and its interpretation.
>
> We believe it unfair to compare our experiments to those appearing in [1]. As we explain in the global response, our work is part of a line of research aimed at achieving a better analytical understanding of deep learning. Given the current state of theory, simplifying assumptions are typically necessary to obtain exact analytical results. Given these constraints, it is not fair to compare the standard transformer used in practice to our simplified model. Real-world applications would anyways be out of scope for our work. We direct the reviewer to our global rebuttal, in which we explain how, when seen in the right context, our work presents novel theoretical insights, as well as several advancements compared to the simplifying assumptions of other closely related theoretical works.
>
> We hope our response clarifies the scope of this work, and highlights our true contributions on both advancing the frontiers of the theoretical research on transformer-like models, and deep learning theory in general.  In our view, the current rating is unfair in light of these contributions. If you find our response convincing/useful, please consider amending the score. Thank you very much.
>
> [1] Liu et al., ICML 2023. Deja Vu: Contextual Sparsity for Efficient LLMs at Inference Time

---

> > ### Comment · Reviewer_5PZg · 2024-08-11
> >
> > Thank you for the response. It is acceptable that this work is theory-focused. Then, the contribution should be Section 3. I have the following questions since I hope the theoretical part to be novel.
> > 1. I roughly checked Appendix C. I do not think the analysis of Transformer architecture is special here. The references [23, 26] mentioned in line 131 are also for linear networks rather than Transformers. My concern is that the analysis here oversimplifies Transformer models into linear models, which weakens the theoretical novelty.
> > 2. What are the theoretical challenges you overcome in Section 3? In other words, why existing methods cannot solve such a problem?
> > 3. The presentation of Section 3 can be improved. It is better to present equations of Section 3 into Theorems/Propositions/Lemmas. Otherwise, it is difficult to find the point in this section.
> > 4. What is the practical significance of this work in real-world applications?

---

> ### Author Response · Authors · 2024-08-12
>
> Thank you very much for taking the time to reassess the value of our work based on our theoretical results. We would like to address your specific questions below. For additional details, please also refer to our general response, where we provide an overview of our contributions, compared to previous comparable works.
>
> 2. *What are the theoretical challenges you overcome in Section 3? In other words, why existing methods cannot solve such a problem?*
>
> Currently, works aiming at providing analytical results for learning in transformers need to consider simplified architectures and/or training regimes (cf. Introduction). The methods adopted to study such simplified models are typically well established (e.g. NNs as Gaussian Processes equivalence, teacher-student setting, dynamic mean-field theory, BPKR, etc...). The challenge, however, is to identify simplified models that are analytically tractable, yet rich enough to offer insights related to specific features of transformers.
>
> In this landscape, the challenges we overcome are:
>
> - a. To consider a model beyond a single-head and/or single-layer architecture, allowing for the existence of attention paths—present in large number in actual transformers—and the characterization of their interplay.
>
> - b. To characterize such a model beyond the Gaussian Process (GP) limit, in which the hidden weights remain random after learning, and therefore cannot learn any data-dependent structure, in particular any clever combination of attention paths.
>
> By including the above features, our analysis reveals a previously uncovered mechanism of attention paths interplay, implemented by learned structures in the value weights, which are predicted and described in our theory by the order parameter. As noted by both reviewers 9bWs and Y9xT, these data-dependent structures remained inaccessible to previous studies which, even when considering multi-layer multi-head transformers, only did so in the GP limit.
>
> 1. *I roughly checked Appendix C. I do not think the analysis of Transformer architecture is special here. The references [23, 26] mentioned in line 131 are also for linear networks rather than Transformers. My concern is that the analysis here oversimplifies Transformer models into linear models, which weakens the theoretical novelty.*
>
> The model we consider is not linear. It is linear in the value weights, but highly nonlinear in the query-key weights, through the attention operation. As we explain in the last part of Section 2, the model can be seen as a deep linear network in the value weights, which is however applied to a highly nonlinearly expanded input of dimension $N_0 H^L$: it reads from the $H^L$ attentioned inputs (i.e. the input transformed by a specific attention path), which are nonlinear functions of the original input of size $N_0$.
>
> A more precise statement is that we exclusively characterize the learning of those weights in which the network is linear, i.e. the value weights (in this sense applying the BPKR technique found in [23, 26]). The other query-key weights, however, are still learned through gradient descent; simply, we do not attempt to describe their learning mechanism in our theory.
>
> The value weights learn to combine architectural features specific to the transformer—the nonlinear attention paths—showcasing a mechanism of attention paths combination that is specific to this work. Also note that the learning of the linear value weights is itself highly nonlinear: this is what gives rise to the nontrivial combination mechanism, which better aligns the network kernel with the task and improves generalization, as shown by our theory and experiments.
>
> 4. *What is the practical significance of this work in real-world applications?*
>
> As a theory-focused research, we admit that the immediate implications of our results to practical applications are limited. Nevertheless, we'd like to believe that our work is a progress toward a more practically useful theory. Our order parameter is a simple measure of the learned weights, which provides useful information about the role and interplay of different attention paths. While work is still needed to extend its application to state-of-the-art networks, we illustrated its efficacy in our head pruning experiment (Sec 4.2.1) connecting our theoretical results in the Bayesian setting to more practical models trained with gradient descent.

---

> > ### Author Response · Authors · 2024-08-12
> >
> > 3. *The presentation of Section 3 can be improved. It is better to present equations of Section 3 into Theorems/Propositions/Lemmas. Otherwise, it is difficult to find the point in this section.*
> >
> > We thank you for your feedback, which helps us improve the communication of our findings. Currently the manuscript alternates the presentation of our results with their interpretation and implications for generalization performance. The revised manuscript will be reorganized to contain a paragraph with a formal theory statement, containing definitions, assumptions and results. The interpretation and discussion of such results will be given after such a statement. The statement is provided in the next comment.
> > Please refer to the submitted manuscript for any referenced equations which do not appear in the statement itself (i.e. Eqs 1-8, which will appear in the manuscript before the theory statement). Also note that we made the following changes in notation:
> > - 1. We renamed the query and key weights from $\textbraceleft Q ^{\left(\ell\right)h},K{} ^{\left(\ell\right)h}\textbraceright _{\ell,h=1} ^{L,H}$ to $\textbraceleft W _{Q} ^{\left(\ell\right)h},W _{K} ^{\left(\ell\right)h}\textbraceright _{\ell,h=1} ^{L,H}$, in order to avoid confusion with the kernel $K$.
> > - 2. We renamed the network predictor on a test example from $f ^{P+1}$ to $f ^{*}$.
> > - 3. We renamed the network inputs from $x ^{\left(0\right)}$ to $x$.
> >
> > This new naming scheme will be consistently implemented throughout the revised manuscript.
> >
> > As a minor note on terminology, please note that this work uses the approach of theoretical physics, which has a long-standing tradition in developing theories of neural networks. Following this approach, we provide our analytical results in the form of Results and Derivations, rather than Theorems and Proofs.

---

> > > ### Author Response · Authors · 2024-08-12
> > >
> > > ### Theory statement ###
> > >
> > > **Definitions.** Consider a training dataset consisting of $P$
> > > inputs $x ^{\mu}\in\mathbb{R} ^{N _0\times T}$ and associated labels
> > > $y ^{\mu}\in\mathbb{R}$, where $\mu=1,\ldots P$. Call $X\coloneqq \textbraceleft x ^{\mu} \textbraceright _{\mu=1} ^{P}$
> > > the set of training inputs and $Y\in\mathbb{R} ^{P}$ the vector of
> > > training labels with $\mu$-th component $y ^{\mu}$. Consider a network
> > > defined by Eqs. (1-4) and in particular call $f ^*$ the network
> > > output (Eq. 4) corresponding to a test input $x ^* \in\mathbb{R} ^{N _0\times T}$.
> > > We remind the reader of the following network hyperparameters: the
> > > input's embedding dimension $N _0$, the hidden layer's width $N$,
> > > the number of tokens $T$, the number of attention heads per layer
> > > $H$, and the number of attention layers $L$.
> > >
> > > **Assumptions.** Assume the network weights $\Theta\coloneqq\left(V ^{\left(0\right)},\textbraceleft V ^{\left(\ell\right)h}\textbraceright  _{\ell,h=1} ^{L,H},a\right)$
> > > are distributed according to the Bayesian posterior distribution defined
> > > in Eq. 8, with temperature $\mathcal{T}>0$, while the query and key
> > > weights $\textbraceleft W _{Q} ^{\left(\ell\right)h},W _{K} ^{\left(\ell\right)h}\textbraceright  _{\ell,h=1} ^{L,H}$
> > > are fixed.
> > >
> > > Assume $N,N _0,P\to\infty$, with $P/N\coloneqq\alpha\in\mathbb{R} ^{+}$
> > > and $P/(N _0H ^{L})\coloneqq\alpha _0\in\mathbb{R} ^{+}$, where $\alpha$,
> > > $\alpha _0$ as well as other size parameters $T,H,L\in\mathbb{\mathbb{N}}$
> > > are finite.
> > >
> > > **Results.** Under the above assumptions,
> > >
> > > (1) the mean predictor under the posterior distribution (Eq. 8) is
> > > given by
> > > $$\mathbb{E}\left[f ^{*}\right]=k ^{\top}\cdot\left(K+\mathcal{T}\mathbb{I}\right) ^{-1}Y, \qquad (9) $$
> > > where the average is w.r.t. to the posterior distribution (Eq. 8).
> > >
> > > The vector $k\in\mathbb{R} ^{P\times1}$ and the matrix $K\in\mathbb{R} ^{P\times P}$
> > > are defined in terms of a kernel function $\mathcal{K}:\mathbb{R} ^{N _0\times T}\times\mathbb{R} ^{N _0\times T}\to\mathbb{R}$
> > > as $k ^{\mu}\coloneqq\mathcal{K}\left(x ^{*},x ^{\mu}|U\right)$ and
> > > $K ^{\mu\nu}\coloneqq\mathcal{K}\left(x ^{\mu},x ^{\nu}|U\right)$, for
> > > $\mu,\nu=1,\dots,P$. The kernel function is given by
> > > \begin{equation}
> > > \mathcal{K}\left(x,x'|U\right)=\frac{1}{H ^{L}}\sum _{\pi,\pi'\in\Pi}U ^{\pi\pi'}C _{\pi\pi'}\qquad\mathrm{with}\qquad C _{\pi\pi'}\coloneqq\frac{1}{N _0}\xi ^{\pi}\left(x\right) ^{\top}\cdot\xi ^{\pi'}\left(x'\right)\, \qquad (10)
> > > \end{equation}
> > > where $\xi ^{\pi}\left(x\right)$ is the ``attentioned input'' corresponding
> > > to an input $x\in\mathbb{R} ^{N _0\times T}$, along path $\pi\in\Pi$
> > > (Eq. 7) and $\Pi$ is the set of all attention paths for a given architecture
> > > with $|\Pi|=H ^{L}$.
> > >
> > > The matrix $U\in\mathbb{R} ^{H ^{L}\times H ^{L}}$, called *order
> > > parameter*, is a positive semi-definite matrix given by
> > > \begin{equation}
> > > U=\underset{\tilde{U}}{\mathrm{argmin}} \ S(\tilde{U};X,Y)\, \qquad (11)
> > > \end{equation}
> > > where the scalar function $S$ called the *action* is defined
> > > as
> > > \begin{equation}
> > > S(U;X,Y)=\mathcal{L}(U)+\alpha\mathcal{E}(U;X,Y)\, \qquad (12)
> > > \end{equation}
> > > The scalar function $\mathcal{\mathcal{E}}$, which we call the *energy*,
> > > is given by
> > > \begin{equation}
> > > \mathcal{E}(U;X,Y)=\frac{1}{P}\ln\det\left(K(X,X|U)+\mathcal{T}\mathbb{I}\right)+\frac{1}{P}Y ^{\top}\cdot\left(K(X,X|U)+\mathcal{T}\mathbb{I}\right) ^{-1}\cdot Y, \qquad (13)
> > > \end{equation}
> > > where $K\coloneqq K(X,X|U)$ is the $P\times P$ training kernel matrix,
> > > defined according to Eq. (10). The expression for the scalar function
> > > $\mathcal{L}$, which we call *entropy*, is lengthy and is given
> > > in Appendix B.1.
> > >
> > > (2) In the particular case of a single head per layer $H=1$, $U$ is a scalar, and the entropy assumes the simple form $\mathcal{L}\left(U\right)=\sigma ^{-2\left(L+1\right)}U-\ln\left(U\right)$,
> > > where $\sigma ^{2}$ is the variance of the Gaussian prior on the network
> > > weights $\Theta$ (see Eq. 8).
> > >
> > > (3) For general $H$, $\mathcal{L}\left(U\right)$ is minimized by
> > > $U ^{\pi\pi'}=\sigma ^{2\left(L+1\right)}\delta _{\pi,\pi'}$,
> > > which therefore is always the solution of Eq. (11) in
> > > the GP limit defined by $\alpha\to0 ^{+}$.
> > >
> > > (4) The matrix $U$ obeys the following relation
> > > \begin{equation}
> > > U ^{\pi\pi'}=\frac{1}{N}\mathbb{E}[V _{\text{eff}} ^{\pi}\cdot V _{\text{eff}} ^{\pi'\top}], \qquad (14)
> > > \end{equation}
> > > where $V _{\text{eff}} ^{\pi}\in\mathbb{R} ^{1\times N}$ are the effective weights along path $\pi$ (Eq. 6).
> > >
> > > **Derivation:** See Appendix C.

---

> > > > ### Comment · Reviewer_5PZg · 2024-08-13
> > > >
> > > > Dear authors,
> > > >
> > > > Sorry for the late reply. I think using techniques from linear networks [23,26] to derive theoretical results in Transformers is risky when the network goes deep and becomes more nonlinear. Equations 11 and 12 explain that when $\alpha\rightarrow 0$, only the same-path kernels are used, while when $\alpha>0$, many cross-path kernels are used, which are verified in Figure 2(e). What if the model becomes deeper? The 3-layer experiments in Figure 6 do not include such results as Figure 2(e). If time allows, it is better to try 12-layer Transformers or 6-layer Transformers, since they are the size of GPT-2 base/small and are deep enough, which can help to address my concern.

---

> ### Author Response · Authors · 2024-08-13
>
> Thank you for your reply. We would like to address your concerns and some misunderstandings.
>
> As we explained in our previous comment, the model we consider is linear in the value weights, while it is nonlinear in the query-key weights (i.e. the attention paths). Since we apply the BPKR technique only to characterize the learning of the value weights, we are not making any approximation and our results are *exact* at any depth for the model under consideration. There is no “risk” of the theory breaking at larger depths since the network will always stay linear in the value weights, at any depth.
>
> We also explained in our first rebuttal that simulating networks of larger depth is prohibitive in the Bayesian framework, but also out-of-scope for this theory-focused work since the theory is exact at any depth, and the experiments have the main purpose of illustrating its mechanisms in a minimal setting. We can however promise to attempt to simulate as-deep-as-possible networks for the appendix of the final version of the manuscript, though there is no reason to expect different results for the model under consideration.
>
> A related question raised by Reviewer Y9xT that we find relevant here is whether our theory could be extended to a network that is nonlinear in the value weights, which we do not consider in this work. This is an interesting mathematical challenge, which future work could tackle by building upon our theoretical advancements (as discussed in our Discussion, and in our dialogue with reviewer Y9xT). We kindly ask the reviewer to evaluate our contributions in terms of these theoretical advancements, which have been highlighted in our global response, as well as by reviewers Y9xT and 9bWs.  While we openly admit that deep learning theory is still far from analytically characterizing state-of-the-art transformers, we make several important steps forward compared to other related works.
>
> Regarding the 3-layer network in Figure 6, we did not show the order parameter simply because the objective of the figure was different. We have the measurement of the order parameter for that network and, unsurprisingly given the exactness of the theory, we clearly see that for large alpha it contains strong off-diagonals, while for small alpha it is diagonal. This makes sense as the order parameter needs to account for the boost in performance for larger alpha which is shown in Figure 6. We will include these plots of the order parameter in figure 6.

---

### Author Rebuttal · Authors · 2024-08-06

We thank the reviewers for their valuable time and their constructive comments. All reviewers agree on two main points:
- Our theoretical contribution is “interesting” (15FG and 5PZg), “novel” (Y9xT and 9bWs) and “technically sound” (Y9xT).
- Experiments thoroughly validate our theoretical results.

However, some reviewers raised concerns about the scale of the experiments (5PZg, 15FG) and the simplified architecture compared to the standard transformer (Y9xT, 15FG).
While we acknowledge that our theoretical results involve several simplifications, we believe these are outweighed by the paper's novel contributions, especially when considered within the appropriate context and scope of this theory-focused work (as noted by 9bWs and Y9xT).

In this global response, we aim at better clarifying the context and scope of this work, and highlight important contributions currently overlooked in the reviews by 15FG and 5PZg.

We would first like to emphasize that our work is part of a line of research aimed at achieving a better analytical understanding of deep learning. It is important to underline that the focus is on obtaining *exact analytical results*—in our case, an exact expression for the network's predictions on test examples—rather than on applications. In particular, it is unfair to compare our results with those from heuristic and phenomenological approaches, which, even when inspired by theory, are less constrained by rigor and free to focus on any arbitrary architecture used in practice.

Given the current state of the art in exact analytical methods, simplifying assumptions are typically necessary to obtain analytical results. In this context, we believe our work provides considerable contributions in the two following directions:
1. **Advances in the analytical understanding of learning in transformer-like architectures.** While from an application-focused perspective, our model still appears simplified with respect to the standard transformer, from the perspective of theory it integrates several key features that were not considered in previous works. These are the combination of (cf. Sec. 1 “Introduction” for references):
    - a. **A multi-head, multi-layer architecture.** Most previous works either considered single-head or single-layer models, which did not allow for a characterization of the interplay between attention paths.
    - b. **A large number of training examples.** Previous works considering the standard transformer architecture typically do so in the Gaussian Process (GP) limit, in which the network width $N→\infty$ while the number of examples $P$ stays finite (i.e. $P<<N$ in practice). Here instead we consider the more realistic case of $P=\alpha N$ for finite $\alpha$. As shown in our work, this is fundamental to uncover interesting learning mechanisms, which are lost in the GP limit.
    - c. **An exact derivation of the network predictor, independent of assumptions on the training data.** Even when including one of the features (a) or (b), previous works either provided results in the form of generalization bounds rather than exact expressions for the network predictor, and/or required specifying a model for the training data statistics. In contrast, our results apply to any realistic training dataset.

    While we openly acknowledge that deep learning theory has still a long way to go to analytically understand transformers, we believe to have made important steps forward in these still underdeveloped directions. Importantly, we believe that what we lose by making certain simplifications in our model is outweighed by what we gain in terms of the novel features described above. For example, note that the simplification of feeding the attention heads with the bare input (which is of concern for reviewers Y9xT and 15FG) is what allows us to analytically tackle a multi-head multi-layer architecture, thereby uncovering the mechanism of attention paths interplay. Therefore, while surely a limitation, this assumption allows us to make a significant step forward, on which future work could build upon, for example working on relaxing our assumption.
2. **Advances in the theory of deep learning in general.** While we focused on a transformer-like architecture due to its relevance to recent advancements in AI, our results have broader implications for the theory of deep learning. As mentioned above, exact derivations of the network predictor without assumptions on the dataset structure are typically limited to the network’s Gaussian Process (GP) limit. However, as shown in [Li, Sompolinsky, PRX 2021], it is only by going beyond this limit that the network’s hidden weights learn data dependencies. In the original deep linear network considered in [Li, Sompolinsky, PRX 2021], this phenomenon did not affect the mean predictor and only appeared as a scalar renormalization of the predictor’s variance. As correctly noted by Reviewer Y9xT, our work provides an analytically tractable yet reasonably powerful model in which the hidden weights learn a data-dependent structure that significantly enhances the performance of the mean predictor, which is now controlled by a matrix order parameter. There are currently only a few examples of such models in the literature [Li, Sompolinsky, NeurIPS 2022; Aiudi, et al., arXiv:2307.11807]. Furthermore our work provides a novel understanding of the order parameter as aimed at maximizing kernel-task alignment (the negative log-likelihood argument), allowing for a more direct interpretation of the renormalization phenomenon. In short, we believe our work makes an important contribution toward uncovering interesting learning mechanisms that can only emerge with a large number of training examples.

In synthesis, our work provides significant contributions to a theory-focused line of research that has always been of high relevance to NeurIPS, and which we hope to have better contextualized in our reply.

---

### Decision · Program_Chairs · 2024-09-25

**Decision:**

Accept (poster)

**Comment:**

The paper studies thermodynamic limits of Transformers in a regime where the key-query matrices are fixed to some pre-trained forms. This leads to interesting new data-dependent kernels that exploit the structure of "attention paths" in order to show generalization benefits. The reviewers found this viewpoint interesting and relevant for understanding the benefits of Transformers.